# Fast and robust drift correction for single-molecule localization microscopy

Mengdi Hou[1,4], Jianyu Yang[1], Mingjie Yang[1], Fen Hu[1], Rongge Zhao[1], Yuhang Pan[1], Wan Li[2], Mingxin Chen[1], Jingjun Xu [1], Ke Xu [2] ✉ & Leiting Pan [1,3] ✉

Owing to its gradual accumulation of molecular positions, single-molecule localization microscopy (SMLM) depends on the proper correction of sample drifts that occur during data acquisition. However, current data-based drift-correction approaches for SMLM are often unreliable and time-consuming, limiting the achieved resolution and throughput. Here we report nearest paired cloud (NP-Cloud), a fast and robust SMLM drift-correction method. By pairing the nearest molecules in SMLM data segments and calculating their displacements within a small search radius, NP-Cloud efficiently utilizes the continuously valued positions of each super-localized molecule while drastically reducing the computational cost. With both simulated and experimental SMLM data, we thus demonstrate substantially improved robustness and fidelity for drift correction in three dimensions, as well as speeds >100-fold faster over traditional single-referenced approaches and >$10^4$ faster over traditional cross-referenced redundant approaches. Excellent drift corrections are achieved for diverse samples within seconds. We thus provide a robust, fast, and practical solution to SMLM drift correction.

Recent years have witnessed the fast growth of super-resolution microscopy (SRM) methods based on the mass accumulation of single-molecule positions, i.e., single-molecule localization microscopy (SMLM)[1–4]. In SMLM, in each camera frame, tens of single emitting molecules, as random subsets of the total fluorescent labels, are localized to ~10 nm spatial precision across the wide field. Fluorescence on-off switching then stochastically samples different labeled molecules over many camera frames to gradually build up a super-resolved image.

Distinct from scanning-based microscopy methods, in SMLM each raw camera frame provides partial information of the entire field of view, and one often accumulates single-molecule positions over >$10^4$ camera frames to achieve high-density coverage of the underlying biological structures. Sample drifts in the lengthy data collection process thus lead to global smearing of the entire field in the final super-resolution image. Although it is possible to physically

compensate for drifts during SMLM data acquisition[5,6], the introduction of mechanical components incurs technical challenges and uncertainties. Instead, it is more practical to computationally correct for drift in the data. Whereas initial SMLM experiments used fiducial markers like fluorescent beads to help track sample drifts for post-experiment correction[7,8], it was soon demonstrated that this cumbersome approach can be circumvented by determining drifts based on the collected SMLM data itself[9–11].

SMLM data-based drift correction usually starts with segmenting the entire dataset by frames and then calculating the spatial shifts between the segments. In typical approaches[9–13], the localized single-molecule positions in each segment are spatially binned with a preset grid size, and then cross-correlations are calculated for the pixelated super-resolution images, e.g., via fast Fourier transform (FFT). While conceptually simple, spatial binning in the first step creates potential issues: Large (e.g., >30 nm) grid sizes do not fully utilize the super-

[1]Key Laboratory of Weak-Light Nonlinear Photonics, Ministry of Education, TEDA Institute of Applied Physics and School of Physics, Nankai University, Tianjin, China. [2]Department of Chemistry, University of California, Berkeley, California, USA. [3]State Key Laboratory of Medicinal Chemical Biology, Academy for Advanced Interdisciplinary Studies, Nankai University, Tianjin, China. [4]Present address: The 27th Research Institute of China Electronics Technology Group Corporation, Zhengzhou, China. ✉e-mail: xuk@berkeley.edu; plt@nankai.edu.cn

localization precision, whereas small grid sizes lead to low counts in each bin and deteriorated correlation. Working with the distances between molecules overcomes this issue[14–16], with recent developments demonstrating improved drift correction by calculating nearest-neighbor distances[15,17] or extracting pairwise displacements[16] between the segmented datasets. However, it remains a challenge to obtain robust, optimal drift correction with reasonable computational speeds. Moreover, trial and error with varied settings is often necessary before acceptable outcomes are (hopefully) realized, creating frustration for the SMLM user and limiting the achieved throughput and spatial resolution.

Here we report nearest-paired cloud (NP-Cloud), which provides a robust, efficient, and user-friendly solution to SMLM drift correction. By pairing the nearest molecules in SMLM data segments within a small search radius and calculating their displacements, NP-Cloud efficiently utilizes the super-localized positions of each molecule while drastically reducing the computational cost. With both simulated and experimental SMLM data, we thus demonstrate substantially improved robustness, fidelity, and speed for drift correction in three dimensions. An open-source MATLAB code is further provided with ample example data to make our algorithm available to all SMLM users and code developers.

## Results and discussion

We start by explaining the basic algorithms of NP-Cloud and demonstrating with simulated data how it extracts the spatial shift between two sets of single-molecule localizations. Starting with the zero-shift condition, we simulated two datasets (drawn as blue vs. red in the left panel of Fig. 1a) with initially identical single-molecule positions, and then added random $xy$ scattering of standard deviations $\sigma = 10$ nm to each molecule, to simulate the typical localization uncertainties in SMLM techniques such as STORM[7,9,18].

For every localization in the red channel, we searched for its nearest-paired (NP) localization (nearest neighbor) in the blue channel within a search radius, and calculated a vectorial displacement between the paired positions. The resulting vectorial displacements for all localizations were pooled and plotted as $\Delta x$ and $\Delta y$ in two dimensions (right panel of Fig. 1a). We have recently employed a related approach to analyze single-molecule motion across tandem camera frames for single-molecule displacement mapping (SM$d$M)[19–21]. However, here we instead searched for matched molecules between two SMLM datasets, e.g., two segments from an SMLM experiment, to determine relative shifts at the nanoscale. For the scenario in Fig. 1a, the resultant NP displacements appeared as a "cloud" centered at the origin consistent with a 2D Gaussian distribution of $\sigma = 14$ nm, expected from the convolution of the two $\sigma = 10$ nm distributions from the two channels (scaling by the square root of 2). Averaging the coordinates of the NP displacements gave near-zero values of (0.21 nm, 0.16 nm) in $(x, y)$, indicating no global shifts between the two channels.

We next added a small spatial shift between the two channels. We note that as drift occurs gradually during SMLM data collection, spatial shifts between successive segments are normally substantially smaller than the final SMLM resolution (~20 nm) -otherwise, the uncorrected drifts *within* each segment would be similarly large, defeating the purpose of super-resolution microscopy. Consequently, in our drift-correction implementation (below), as we processed the segments sequentially, for each segment, we started by pre-shifting the data using the already calculated drift of the prior segment. NP-Cloud thus only needed to determine the new drift between the current and previous segments and search for nearest positions between the two channels within a small radius, e.g., 50 nm.

Figure 1b presents a case in which the single-molecule positions in the red channel of Fig. 1a were globally shifted by (+20 nm, +10 nm) in $(x, y)$. NP analysis showed a commensurate shift of the center of the resultant NP-Cloud (right panel of Fig. 1b), suggesting the feasibility of extracting spatial shifts based on this analysis. Meanwhile, Fig. 1c presents a case in which no spatial shifts existed between the two channels, but 9-fold more random single-molecule localizations were added to each channel, so that for each channel, only 10% of the localizations correlated with the other channel. NP analysis now showed a uniform background together with the high-density cloud centered at the origin.

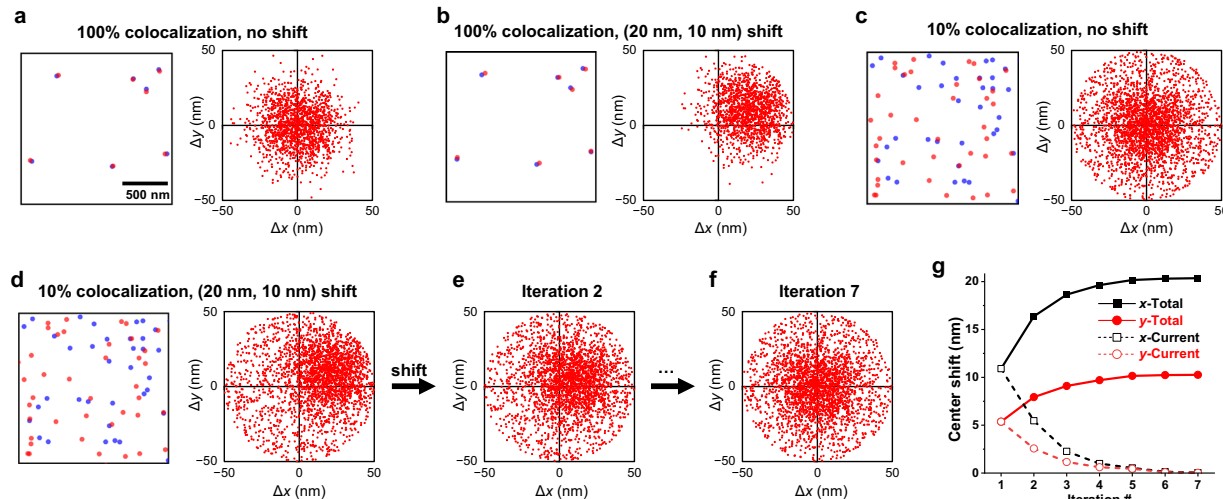

**Fig. 1 | NP-Cloud extracts nanoscale spatial shifts between two sets of single-molecule localizations. a** Left: Zoom-in view of a small region of simulated single-molecule localizations in two channels (red and blue) with initially identical single-molecule positions but added Gaussian scattering of standard deviations $\sigma = 10$ nm to each localization. Right: Corresponding nearest-paired (NP) analysis result: The 1600 localizations in the red channel across the entire field were each searched for their nearest paired localization in the blue channel with a search radius of 50 nm, and the calculated vectorial distances (displacements) were plotted as $\Delta x$ and $\Delta y$ in 2D. **b** Left: Simulated data as (**a**), but with all the single-molecule positions in the red channel uniformly shifted by (+20 nm, +10 nm). Right: Corresponding NP-analysis result. **c** Left: Simulated data as (**a**), but with 9-fold additional random single-molecule localizations added to each channel. Right: Corresponding NP-analysis result. **d** Left: Simulated data as (**c**), but with all the single-molecule positions in the red channel uniformly shifted by (+20 nm, +10 nm). Right: Corresponding NP-analysis result. **e** NP-analysis result of Iteration 2, after shifting the localizations in the red channel by the averaged displacements of the NP-analysis results in **d**. **f** NP-analysis result after 5 more iterations of shifting and NP analysis. **g** Stepwise (open symbols) and total (filled symbols) shifts in $xy$ through the above iteration process.

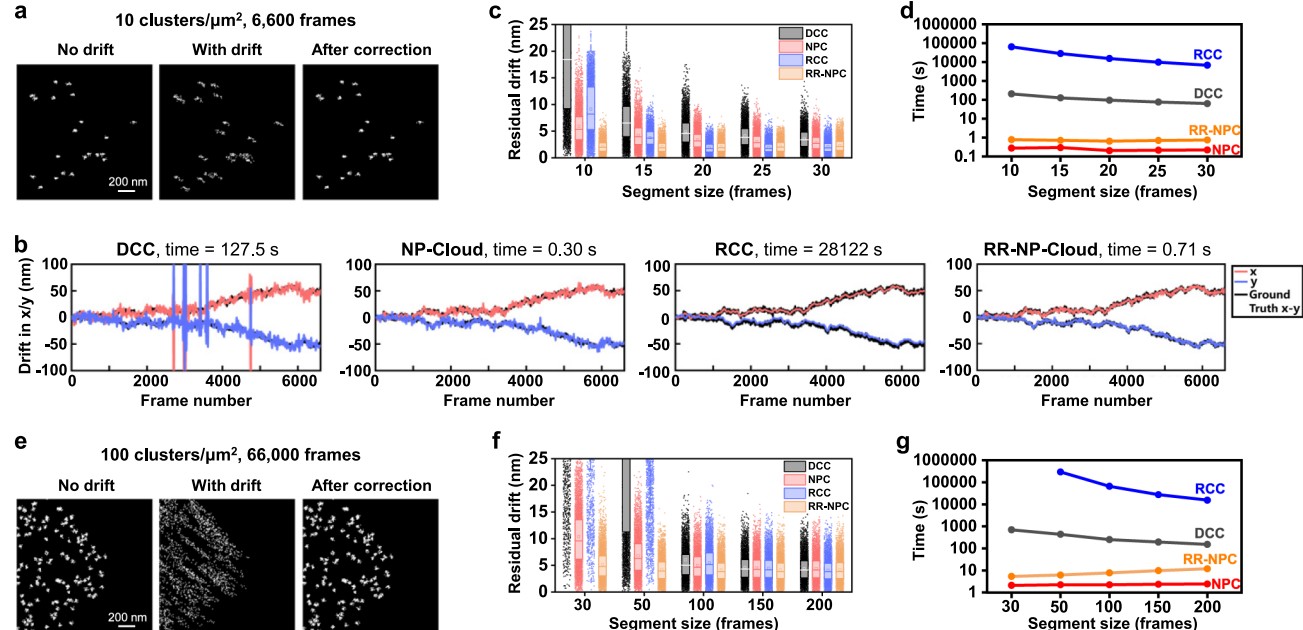

**Fig. 2 | Comparison of drift-correction results on simulated SMLM data with direct cross-correlation (DCC), nearest paired cloud (NP-Cloud), redundant cross-correlation (RCC), and resampled-reference NP-Cloud (RR-NP Cloud).** **a** Zoom-in view of a small region of the simulated SMLM dataset of 6,600 frames with relatively sparse features of 10 clusters/μm², without drift (left), after the addition of drift (center), and after drift correction by RR-NP Cloud (right). **b** Drift correction curves in $x$ (red) and $y$ (blue) based on the 4 different methods, versus the ground truth (black), for the same segment size of 15 frames/segment. **c** Distributions of residual drift in the $n = 6600$ frames, calculated as the absolute distance between the estimated drift and the ground truth, for the four methods at different segment sizes. Dots: residuals of individual frames; box boundaries: the first and third quartiles; center lines: medians; small squares: means; whiskers: SD. **d** Corresponding time spent under the different conditions. **e** Zoom-in for a small region of another set of simulated SMLM data of 66,000 frames with denser features of 100 clusters/μm². **f** Distributions of residual drifts of $n = 5000$ randomly sampled frames from the drift correction results of this dataset, for the four methods at different segment sizes. Box plot was constructed in the same way as (**c**). **g** Corresponding time spent under the different conditions. Note: For the segment size of 30 frames/segment, RCC was unable to complete after >10 days, and so is omitted in **f, g**.

Figure 1d combines the above spatial shift and uncorrelated localizations. NP analysis showed a high-density cloud centered away from the origin representing the shift between the two channels, on top of a uniform background from the uncorrelated localizations. As the background was centered at the origin rather than the center of the NP-Cloud and was further cut off by the search radius, simply averaging the coordinates of all the NP displacements in Fig. 1d underestimates the actual shift of the NP-Cloud center from the origin. To overcome this issue, we implemented an iterative algorithm, in which the above underestimated spatial-shift values were used to shift the red channel closer to the blue channel, so the shifted data could go through NP analysis again, now with a smaller remaining shift between the two channels and hence a less asymmetric background in the NP-analysis result (Fig. 1e). Coordinate averaging was then used again to find out the residual shifts between the two channels after the first shift. The above process was repeated for a few rounds until convergence (Fig. 1f), when the calculated mean position of the cloud after the shift no longer moved closer to the origin when compared to the previous iteration. Residual shift was thus minimized between the two channels, and the NP-Cloud (Fig. 1f) appeared near-identical to that without shift (Fig. 1c) with no detectable asymmetric backgrounds. Thus, this stepwise correction process asymptotically shifted the NP-Cloud center to the origin based on the updated position in each iteration. The total shift after reaching convergence (+20.4 nm, +10.3 nm) (Fig. 1g) gave an estimation of the initial displacement between the two channels, which was in good agreement with the ground truth (+20 nm, +10 nm).

Thus, we have shown that NP-Cloud analysis allowed robust extraction of spatial shifts between two sets of single-molecule localizations in the presence of uncorrelated localizations. This capability offers a useful framework for SMLM data-based drift correction. To

this end, we constructed simulated SMLM data in which clusters of single-molecule localizations were distributed in a continuous region occupying 2/3 of a typical 40 μm × 40 μm camera frame. Each cluster contained ~20 single-molecule localizations that were scattered with standard deviations of 10 nm in both $x$ and $y$ directions, simulating the repeated localization of the same label in SMLM. The simulated localizations were randomly assigned to different camera frames at 30 localizations/frame, considering our STORM data typically has 20–50 single-molecule localizations in each frame (see Data Availability). We further note that since consecutive frames are grouped into segments and taken together to estimate drifts between different segments in drift-correction calculations, the precise count of localizations in each frame is not a critical factor. Molecules were then $xy$-shifted according to their assigned frames, using a preset drift curve measured with beads on a typical SMLM setup (e.g., the "ground truth" curve shown in Fig. 2b and Supplementary Fig. 1a).

For a simulation with relatively sparse features (10 clusters/μm²), the above condition of 20 localizations per cluster was achieved with 6600 frames, thus simulating a relatively short SMLM acquisition (Fig. 2a). For drift correction, we started by segmenting the drift-added data by the frame number using a fixed segment length, e.g., 15 frames/ segment for a total of 440 segments.

For the commonly used direct cross-correlation (DCC) drift correction[9,11], the single-molecule localizations in each segment were first grid-binned with a grid size of 15 nm. The resultant pixelated SMLM images of different segments were then each cross-correlated with the first segment through FFT, and peak positions in the correlation results were taken as the estimated spatial shifts. For the 15 frames/segment condition, we thus noted the frequent occurrence of faulty shift values, shown as spikes in the estimated drift curves deviating from the ground truth (Fig. 2b).

For NP-Cloud-based drift correction, as noted above, with each segment we pre-shifted the single-molecule localizations by the calculated drift of the prior segment (versus the first segment), and then applied NP-Cloud to determine the remaining spatial shift versus the first segment within a small (~50 nm) search radius, using the iterative algorithm introduced above (Fig. 1d–g). This small shift was then added back to the drift value of the prior segment to be recorded as the estimated drift value of the current segment. Note that even though a pre-shift was made based on the drift of the prior segment, correlation was always performed with the first (reference) segment, so errors in the calculated drifts do not accumulate over time.

For the 15 frames/segment condition, NP-Cloud faithfully tracked sample drift for all segments (Fig. 2b). For the even smaller segment size of 10 frames/segment, NP-Cloud still worked robustly as DCC became unworkable (Fig. 2c and Supplementary Fig. 1). While larger segment sizes worked for both methods, finer details in the faster-changing components of the drift were lost (Supplementary Fig. 1). For all segment sizes examined, NP-Cloud consistently provided more faithful drift corrections than DCC when compared to the ground truth (Fig. 2c). These benefits are attributed to the grid-free approach of NP-Cloud, so that all super-localized molecular positions are utilized to the full extent.

It is further important to note that the NP-Cloud-based drift correction is ~400-fold faster than DCC, taking ~0.2 s for each run for the above data (Fig. 2d). The significantly faster speed is attributed to the small search radius needed in NP analysis with our pre-shifting approach, which drastically reduces the computational cost of NP-Cloud. In contrast, in DCC, FFT-based image correlations are calculated for the full frame size regardless of the drift amount.

We next examined another simulated dataset with denser features at 100 clusters/$\mu m^2$. Now the condition of ~20 localizations/cluster was achieved with 66,000 simulated frames, mimicking a longer SMLM acquisition (Fig. 2e). DCC was thus found unworkable for segment sizes of 50 frames/segment or smaller (Fig. 2f and Supplementary Fig. 2). In contrast, NP-Cloud remained robust for different segment sizes down to 30 frames/segment and consistently provided more faithful drift corrections for different segment sizes (Fig. 2f and Supplementary Fig. 2). Timewise, NP-Cloud again outperformed DCC by >100-fold (Fig. 2g).

A notable limitation of DCC and NP-Cloud is that each segment is only compared with the first segment as the reference, which may not contain all structural features of the sample given the stochastic nature of SMLM. Redundant cross-correlation (RCC)[11] improves over DCC (Fig. 2b, c) on this issue by cross-examining the correlations between all segments. However, this treatment drastically increases computational burdens. For the simulated SMLM data of 66,000 frames above, segmentation at 100 frames/segment into 660 segments required ~660/2 = 330-fold more cross-correlations to be calculated in RCC over the single-referenced DCC, with a corresponding increase in the computation time (Fig. 2g).

Whereas NP-Cloud can be similarly implemented in a redundant approach to achieve excellent drift correction ~500-fold faster than RCC (Supplementary Fig. 3), we further report a strategy to circumvent the excessive redundancy in RCC. In this approach, we generated a resampled reference image after a first round of NP-Cloud-based drift correction, by evenly subsampling the drift-corrected single-molecule positions in the entire dataset. This reference image contained substantially more single-molecule localizations (e.g., set to be 12-fold) over the first segment alone, thus capturing more structural features. Against this feature-enhanced resampled reference image, a second round of NP-Cloud-based drift correction was performed for each segment using the original single-molecule localizations to achieve more robust shift estimation. We named this approach resampled-reference NP-Cloud (RR-NP Cloud).

For the simulated dataset of 6,600 frames, RR-NP Cloud achieved comparable drift-correction results as RCC but with better stability for small segment sizes (Fig. 2b, c and Supplementary Fig. 1). Meanwhile, the two-pass calculation approach of RR-NP Cloud only doubles the time over NP-Cloud, >$10^4$ faster than RCC (Fig. 2d). For the simulated dataset of 66,000 frames, RCC at 50 frames/segment required a prohibitively long calculation of ~$3 \times 10^5$ s, or ~3.5 days, and the results did not faithfully track the drift (Fig. 2f, g and Supplementary Fig. 2), expected given the large errors in DCC under the same segment size. In comparison, RR-NP Cloud achieved excellent drift correction within seconds. Moreover, RR-NP Cloud worked well for the even more challenging condition of 30 frames/segment (Fig. 2f, g and Supplementary Fig. 2), for which RCC was entirely unworkable. For segment sizes of >100 frames/segment (hence fewer segments), RCC achieved acceptable drift correction results over $10^4$–$10^5$ s (Fig. 2g) or tens of hours, as RR-NP Cloud consistently outperformed in the final drift-correction fidelity (Fig. 2f), again within seconds (Fig. 2g).

We next applied NP-Cloud and RR-NP Cloud to experimental data. Starting with a sample with well-separated clusters at relatively low densities, namely, tropomodulin in erythrocytes (Fig. 3a, b)[22], we showed that NP-Cloud and RR-NP Cloud achieved robust drift corrections for varied segment sizes down to 30 frames/segment (Fig. 3f, h), whereas DCC and RCC were faulty at 100 frames/segment and failed for segment sizes of 50 frames/segment or lower (Fig. 3e, g). Overlaying the clusters in the SMLM images drift-corrected by the different methods showed smaller full width at half maximum (FWHM) values down to 30 nm for NP-Cloud and RR-NP Cloud versus DCC and RCC (Fig. 3c), suggesting better corrections. Moreover, both NP-Cloud and RR-NP Cloud were completed in seconds, >100-fold faster than DCC and >$10^4$ faster than RCC (Fig. 3d). These results are consistent with those observed above with simulated data.

We next examined a more challenging sample in which high-density structures were not well resolved at the SMLM level, the C-terminus of β-spectrin in erythrocytes (Fig. 3i)[22]. Here, the sample included constantly emitting fluorescent beads as fiducial markers to facilitate the evaluation of the drift-correction results, but all drift corrections were performed with the bead localizations excluded. With the same segment size of 100 frames/segment, DCC failed markedly with scattered drift values (Fig. 3j–l). RCC provided usable drift corrections, but still left substantial residual drifts, as reflected both by the final drift-corrected SMLM images (Fig. 3j, k) and by comparing the drift curves with those separately determined from the fluorescent beads (Fig. 3l). In contrast, NP-Cloud and RR-NP Cloud afforded excellent drift correction: crisp drift-corrected SMLM images were obtained (Fig. 3j), including for fluorescent beads not used in the drift-correction calculation (Fig. 3k), and the drift curves matched well with that independently determined from the beads (Fig. 3l).

Supplementary Figs. 4–7 present additional examples for the application of NP-Cloud and RR-NP Cloud to SMLM data of diverse structures, including phalloidin-labeled actin filaments in fixed macrophages (Supplementary Fig. 4) and immunolabeled microtubules (Supplementary Fig. 5), endoplasmic reticulum (Supplementary Fig. 6), and mitochondria (Supplementary Fig. 7) in fixed COS-7 cells, and we provide all data in the supplement for testing (see Code Availability and Data Availability). NP-Cloud and RR-NP Cloud consistently provided excellent drift corrections in seconds, including for difficult cases that could not be properly corrected with DCC and RCC, as shown with drift-correction curves, corrected final images, and Fourier ring correlation (FRC) analysis[23].

Among these, Supplementary Fig. 6 shows an example in which at 200 frames per segment, DCC initially worked but failed for later parts of the data when the count of localizations in each frame dropped due to photobleaching, a common scenario in SMLM. RR-NP Cloud robustly handled the whole dataset, and worked well even for the

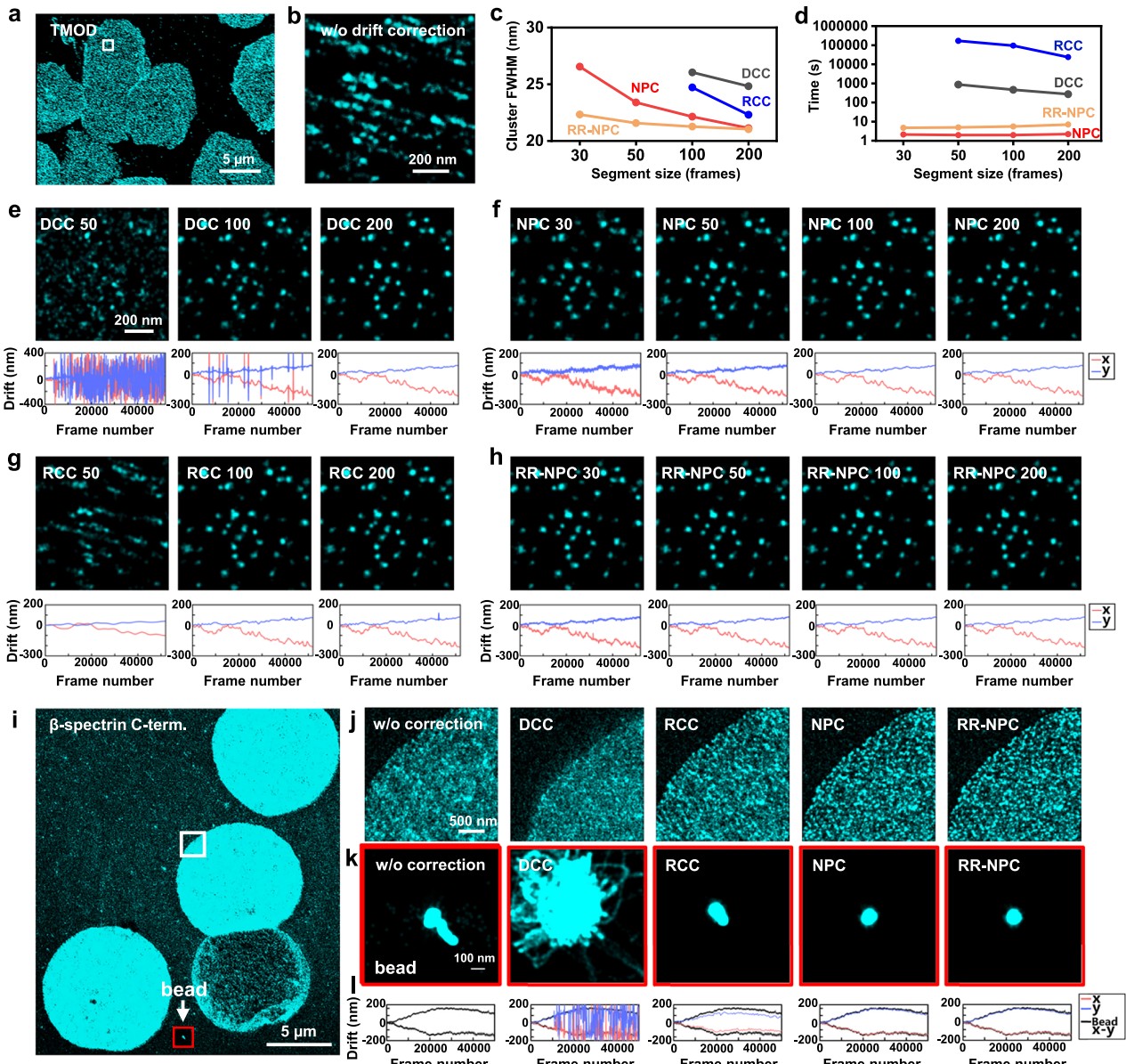

**Fig. 3 | Comparison of drift-correction results on experimental SMLM data with DCC, NP-Cloud, RCC, and RR-NP Cloud. a** An experimental STORM SMLM dataset of immunolabeled tropomodulin in erythrocytes, shown as raw localizations without drift correction. **b** Zoom-in of the box region in **a**. **c** Statistics of the tropomodulin cluster sizes in full width at half maximum (FWHM) for SMLM images drift-corrected by the four methods at different segment sizes, after overlaying the drift-corrected single-molecule localizations of different clusters by their centers. Analysis was not attempted for segment sizes of 50 frames/segment or lower for DCC and RCC as clusters were not well defined. **d** Corresponding time spent under the different conditions. **e** Drift-correction results through DCC at segment sizes of 50, 100, and 200 frames/segment, shown as (top) drift-corrected images for the region in **b** and (bottom) the calculated drifts in *x* (red) and *y* (blue). **f** Drift-correction results through NP-Cloud at segment sizes of 30, 50, 100, and 200 frames/segment. **g** Drift-correction results through RCC at segment sizes of 50,

100, and 200 frames/segment. **h** Drift correction results through RR-NP Cloud at segment sizes of 30, 50, 100, and 200 frames/segment. **i** Another experimental SMLM dataset of immunolabeled C-terminus of β-spectrin in erythrocytes. **j** Zoom-in SMLM images for the boxed region in **i**, before drift correction and after drift corrections by different methods at the same segment size of 100 frames/segment. **k** Corresponding images with and without the drift corrections for a bead in the view that was constantly localized in all camera frames but not used in the drift-correction calculations. **l** Drifts in *x* (red) and *y* (blue) calculated by the different methods, in comparison with the measured bead drifts over different frames (black). Note that we flipped the sign of *y* when drawing the drift curves to reconcile the plotting convention of increasing *y* in the upward direction versus the camera convention of increasing *y* in the downward direction. Data for **a, i** were obtained from at least ten cells across three independent experiments.

much smaller segment size of 50 frames/segment, under which condition DCC completely failed.

Recent years have seen the emergence of new approaches that substantially improved the performance of SMLM data-based drift correction. Wester et al. perform drift correction after brightfield-based registration using nearest neighbor distances limited by a cutoff criterion[15], and provide a MATLAB code package[17]. Running the

provided code with our data, we found that the algorithm, driftCorrectKNN, was similarly fast as RR-NP Cloud but yielded notably larger residual errors and failed for small segment sizes (Supplementary Figs. 8–10). Fazekas et al. estimate drift by calculating the mean shift of all pairwise displacements between segmented datasets[16]. We found this "mean shift" method produced good drift correction results but was excessively slow due to its full redundancy approach as in RCC

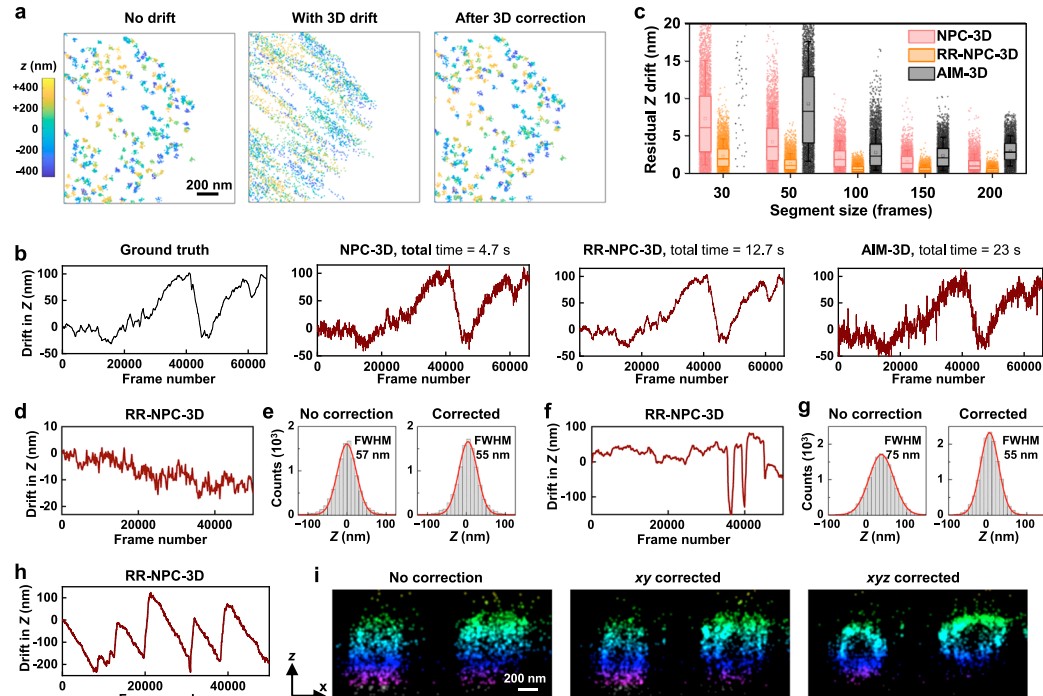

**Fig. 4 | Extending NP-Cloud and RR-NP Cloud to 3D drift correction. a** Zoom-in view of a small region of the simulated 3D-SMLM dataset of 66,000 frames with features of 100 clusters/μm², without drift (left), after the addition of 3D-drift (center), and after drift correction by RR-NP Cloud-3D (right). Color presents the $z$ values. **b** Comparison of the ground-truth $z$ drift curve (black) with those calculated from the simulated 3D-SMLM dataset using NP-Cloud-3D, RR-NP Cloud-3D, and AIM-3D at the same segment size of 50 frames/segment (green). **c** Distributions of residual $z$ drifts of $n$ = 5000 randomly sampled frames from the drift correction results, calculated as the distance between the estimated drift and the ground truth, for the three methods at different segment sizes. Dots: residuals of individual frames; box boundaries: the first and third quartiles; center lines: medians; small squares: means; whiskers: SD. Note that AIM-3D failed for the 30 frames/segment condition and did not generate meaningful drift correction. **d** RR-NP Cloud-3D calculated $z$ drift for the experimental 3D-STORM data of dye-labeled antibody adsorbed to a coverslip, in which $z$ was stabilized by the microscope focus lock. **e** Histograms: Distribution of $z$ values for single-molecule localizations before (left) and after (right) 3D drift correction. Red curves: fits to normal distribution, yielding FWHM of 57 and 55 nm, respectively. **f, g** Similar to **d, e** but with the focus lock disabled, and the focal plane was continuously adjusted manually during imaging. Fits to normal distribution yield FWHM of 75 and 55 nm before and after RR-NP Cloud-3D drift correction. **h** RR-NP Cloud-3D calculated $z$ drift for the experimental 3D-STORM data of immunolabeled mitochondrial outer membrane (TOM20) in a fixed COS-7 cell. When collecting this dataset, the focus lock was disabled, and the sample was allowed to continuously drift in $z$ by ~200 nm each time before the focus was manually adjusted to bring the sample back. **i** Virtual vertical ($xz$) cross sections for a region of the data (marked in Supplementary Fig. 12), without drift correction (left), after only in-plane ($xy$) drift correction through RR-NP Cloud (middle), and after 3D drift correction through RR-NP Cloud-3D (right).

(Supplementary Figs. 8–10). It was ~100× slower than RR-NP Cloud for large segment sizes, and even slower for small segment sizes. For our simulated SMLM data of 66,000 frames, the mean-shift approach took $6 \times 10^5$ s (~7 days) at a segment size of 50 frames per segment, and was unable to complete over 15 days at 30 frames per segment. In contrast, RR-NP Cloud was completed in 10 s for both cases with good results. Most recently, Ma et al. reported adaptive intersection maximization (AIM)[13], which effectively correlates gridded/pixelated SMLM data locally within a small search region and implements a second-pass step to correlate each segment with the full dataset after a first round of drift correction. While these designs drastically reduce computational cost, AIM runs into the same issue as DCC/RCC in its spatially binning of single-molecule localizations, and so fails at small segment sizes (Supplementary Figs. 8–10), under which conditions insufficient localizations are counted in each gridded bin. Thus, RR-NP Cloud provides the most robust drift correction with excellent speed and final results under varied parameters.

We further extended NP-Cloud and RR-NP Cloud to three-dimensional (3D) drift correction. To simulate 3D-SMLM data (Fig. 4a), we generated 66,000 frames of single-molecule localizations with features of 100 clusters/μm² like above, and randomly assigned an initial depth ($z$) value to each cluster in the −400 to +400 nm range. Every localization was then added with a scattered $z$ value of a standard deviation of 22 nm. These $z$-range and standard deviation values are based on what is typically achieved in the cylindrical-lens-based 3D-

STORM[18]. The simulated locations were then shifted in 3D with a preset ground-truth drift trajectory, with the in-plane $xy$ shift being identical to that used above for the simulated 2D data (Supplementary Fig. 2a), and the $z$ shift shown in Fig. 4b.

For drift correction in 3D, the $z$ position of every molecule is considered when matching NP localizations in the reference and comparing datasets to generate the NP-Cloud, using a $z$ search radius of 100 nm given the ~50 nm FWHM of $z$ localization uncertainty. NP-Cloud centers were separately calculated for $xy$ and for $z$. At a segment size of 50 frames/segment, we thus found that NP-Cloud-3D faithfully captured drift in 3D within seconds (Fig. 4a–c and Supplementary Fig. 11). RR-NP Cloud-3D, in which the reference frame was resampled for a second round of drift correction after the first round, further yielded excellent $z$ drift curves indistinguishable from the ground truth (Fig. 4b, c). In comparison, rising methods like AIM[13] yielded substantially larger residual errors in $z$ drift correction (Fig. 4b, c).

We next examined experimental 3D data. For 3D-STORM data acquired with a focus lock, we found that the sample $z$ drifts were typically within <50 nm for the entire dataset (Fig. 4d and Supplementary Fig. 7), under which circumstances $z$ drift correction provided limited improvement to the final results (Fig. 4e for dye-labeled antibody molecules adsorbed to the coverslip and Supplementary Fig. 7 for mitochondria in fixed cells). In contrast, if the focus lock was disabled and one relied on manual focal adjustment to compensate for $z$ drifts during data acquisition, RR-NP Cloud-3D effectively corrected

the degradation in $z$ resolution and generated results similar to that collected with the focus lock on, as demonstrated in Fig. 4f, g for antibody on the coverslip and Fig. 4h, i and Supplementary Fig. 12 for resolving the hollow cross-section of the mitochondrial outer membrane in fixed cells.

In conclusion, we have developed a robust and fast drift correction method based on NP analysis. By pairing the nearest molecules in SMLM data segments and calculating their displacements within a small search radius, NP-Cloud and RR-NP Cloud efficiently utilize the super-localized positions of each molecule to achieve highly robust and accurate drift correction. With the implementation of a pre-shifting approach to limit NP analysis to a small search radius, NP-Cloud also drastically reduced computational cost to achieve >100-fold faster speeds over the traditional DCC method. In a two-pass approach, RR-NP Cloud further resampled the NP-Cloud drift-corrected SMLM data to enhance structural features in the reference frame for a second round of NP-Cloud analysis, thus achieving further improved drift correction by only doubling the computational time over NP-Cloud. Excellent drift corrections were thus achieved for diverse SMLM data within seconds, >$10^4$ faster than the prohibitively time-consuming RCC. A comparison with existing drift-correction methods showed that RR-NP Cloud provides the most robust drift correction with excellent speed. In particular, for reduced segment sizes, which also emulate scenarios in which fewer localizations are made in each frame or otherwise fewer correlated localizations exist between different frames, RR-NP Cloud uniquely provides robust drift correction when other methods fail. Extending the NP-Cloud and RR-NP Cloud approaches to 3D SMLM drift correction also yielded excellent results for both simulated and experimental data, effectively enabling good 3D-SMLM results to be obtained without focus locks. A user-friendly open-source code is provided with ample example data to make our algorithm available to SMLM users and code developers (under Creative Commons Attribution NonCommercial 4.0 International License (CC BY-NC 4.0); see Code Availability and Data Availability). It is further worth noting that except for the segment size, NP-Cloud and RR-NP Cloud require no adjustment of parameters when running on diverse datasets. We thus provide a fast, robust, and practical solution to SMLM data-based drift correction.

## Methods

Codes and test data are provided under Creative Commons Attribution Non Commercial 4.0 International (CC BY-NC 4.0) Licensee at: https://doi.org/10.5281/zenodo.16513620. Coding was done in MATLAB. Computation was performed on a PC computer with Intel Core i7-13700K CPU, 64 G RAM (2400 MHz), and 1 Tb SDD (Samsung 980 Pro).

### Simulations

Single-molecule localizations were simulated to provide data with ground truths. To demonstrate how NP-Cloud extracts spatial shifts between two sets of single-molecule localizations (Fig. 1), single-molecule localizations were first simulated as random locations evenly distributed at 2 molecules/μm² over a typical camera frame size of 40 μm × 40 μm. The same localizations were assigned to two channels (blue vs. red), and then each localization was separately added a random Gaussian scattering of standard deviations $\sigma = 10$ nm in both the $x$ and $y$ directions to simulate single-molecule localization uncertainties (Fig. 1a). A global shift of ($x = +20$ nm, $y = +10$ nm) was then added to all the single-molecule positions in the red channel to generate Fig. 1b. To generate Fig. 1c, 9-fold additional random single-molecule localizations were added to each channel for the unshifted data in Fig. 1a. All the single-molecule localizations in the red channel of Fig. 1c were then globally shifted by ($x = +20$ nm, $y = +10$ nm) to generate Fig. 1d. For simulation of SMLM data in Fig. 2, clusters of single-molecule localizations were distributed in a continuous region occupying 2/3 of a 40 μm × 40 μm camera frame. Each cluster contained ~20 single-

molecule localizations that were scattered with standard deviations of 10 nm in both $x$ and $y$ directions, simulating the repeated localization of the same label in SMLM. The simulated localizations were randomly assigned to different camera frames at 30 localizations/frame. Two scenarios with sparse (10 clusters/μm²) and dense (100 clusters/μm²) features were thus achieved with 6600 and 66,000 simulated frames, respectively, mimicking a short (Fig. 2a) and a long (Fig. 2e) SMLM acquisition. Molecules were then $xy$-shifted according to their assigned frames, using a preset drift curve measured with beads on a typical SMLM setup (below).

### NP-Cloud implementation for extracting spatial shifts

NP-Cloud is based on analyzing the vectorial displacements for the nearest paired (NP) localizations between two sets of single-molecule positions, an approach inspired by single-molecule displacement/diffusivity mapping (SM$d$M)[19,20]. However, in SM$d$M, molecules are paired between tandem frames to detect transient displacements, whereas in NP-Cloud we match molecules between two SMLM datasets, e.g., two consecutive segments from an SMLM dataset, to determine relative shifts at the nanoscale. Moreover, with SM$d$M, the accumulated vectorial displacements are fit for a distribution centered at the origin. In NP-Cloud, we aim to find out the center of the distribution cloud to extract the spatial shift between the two datasets. To achieve this goal, we start by averaging the coordinates of all the NP displacements. This calculation underestimates the spatial shift due to the asymmetric background discussed in the text. To overcome this issue, an iterative converging algorithm was implemented: The underestimated spatial shift from the above coordinate averaging was used to shift the red channel closer to the blue channel; the shifted data then went through NP analysis again, now with a smaller remaining shift between the two channels and hence a less asymmetric background in the NP-analysis result (Fig. 1e). Coordinate averaging was then used again to find out the residual shifts between the two channels after the first shift. The above process was repeated a few rounds until convergence (Fig. 1f), namely, when the calculated mean position of the cloud after the shift was no longer closer to the origin when compared to the previous iteration. Thus, this stepwise correction process asymptotically shifted the NP-Cloud center to the origin based on the updated position in each iteration, and the total shift after reaching convergence gave an estimation of the initial displacement between the two channels.

### NP-Cloud implementation for drift correction of SMLM data

The SMLM data, either simulated or experimental, were first segmented by the frame number using a fixed segment length. For the commonly used DCC drift correction[9,11], the single-molecule localizations in each segment were first grid-binned with a grid size of 15 nm. The resultant pixelated SMLM images of different segments were then each cross-correlated with the first segment through FFT, and peak positions in the correlation results were taken as the estimated spatial shifts. For drift correction based on NP-Cloud, the segments were sequentially processed: With each segment, we pre-shifted the single-molecule localizations by the calculated drift of the prior segment (versus the first segment), and applied NP-Cloud (above) to determine the remaining spatial shift versus the first segment within a small (~50 nm) search radius. This small shift was then added back to the drift value of the prior segment to be recorded as the estimated drift value of the current segment. After all segments were processed, the estimated drift values were linearly interpolated to generate the drift value of each frame, and they were used to shift the single-molecule locations in the whole dataset for drift correction.

### RR-NP Cloud implementation

For RR-NP Cloud, after NP-Cloud calculations as above for all segments, the single-molecule locations were shifted by segments to generate a first round of roughly drift-corrected dataset. This dataset

was then evenly subsampled through all frames to generate a new reference image. This reference image contained substantially more single-molecule positions (e.g., set to be 12-fold) over the first segment to capture more structural features. Against this feature-enhanced resampled reference image, a second round of NP-Cloud-based drift correction was then performed for each segment using the original single-molecule localizations.

## Extension to 3D drift correction

Simulated 3D-SMLM data was generated with 66,000 frames of single-molecule localizations with features of 100 clusters/$\mu m^2$ like above for the simulated dense sample above, but an initial depth ($z$) value was randomly assigned to each cluster in the −400 to +400 nm range. Every localization was then added with a scattered $z$ value of a standard deviation of 22 nm. The simulated locations were then shifted in 3D with a preset ground-truth drift trajectory. 3D drift correction was performed analogously as 2D drift correction above, but the $z$ position of every molecule is considered when matching NP localizations in the reference and comparing datasets to generate the NP-Cloud. A $z$ search radius of 100 nm was used in matching the NP localizations given the typically ~50 nm FWHM of $z$ localization uncertainty. NP-Cloud centers were separately calculated for $xy$ and for $z$ to estimate relative shifts in 3D between the reference and comparing datasets.

## Experimental data

Experimental data were collected on SMLM setups based on Nikon Ti-E inverted fluorescence microscopes[24,25] using a 100× oil-immersion objective (Nikon CFI Plan Apochromat λ). Typical drift curves were obtained using bead samples (Invitrogen, F8783; 1:1000000 dilution). SMLM datasets were obtained under typical STORM protocols[18]. COS-7 cells[26,27] were cultured on uncoated glass coverslips and fixed with 3% paraformaldehyde and 0.1% glutaraldehyde in DPBS for 20 min and then washed with a freshly prepared 0.1% sodium borohydride solution in DPBS, followed by immunofluorescence using anti-tubulin (Sigma, T6199), anti-TOM20 (Proteintech, 11802-1-AP), or anti-Rtn4a/b (ThermoFisher, PA1-41220) primary antibodies and Alexa Fluor 647 (AF647)-conjugated secondary antibodies (Invitrogen, A31571 and A21245). Red blood cells[22] were first adhered to poly-L-lysine-coated glass coverslips, and then fixed in 4% paraformaldehyde in Dulbecco's phosphate-buffered saline (DPBS) for 10 min, followed by immunofluorescence using anti-tropomodulin (OriGene, TA503146) or anti-β-spectrin (C-terminus; NeuroMab, 73-374) primary antibodies and AF647-conjugated secondary antibodies. Macrophages[25] were first fixed and extracted[28] with 0.3% glutaraldehyde and 0.25% Triton X-100 in the cytoskeleton buffer (CB, 10 mM MES, pH 6.1, 150 mM NaCl, 5 mM EGTA, 5 mM glucose and 5 mM $MgCl_2$) for 1 min, and then post-fixed with 2% glutaraldehyde in CB for 15 min, washed with 0.1% sodium borohydride in DPBS, and labeled with AF647-conjugated phalloidin (Invitrogen, A22287). STORM was performed using a Tris-HCl–based imaging buffer containing 5% glucose, 100 mM cysteamine (Sigma-Aldrich, 30070), 0.8 mg/mL glucose oxidase (Sigma-Aldrich, G2133), and 40 μg/mL catalase (Sigma-Aldrich, C30). A 647 nm laser illuminated the sample at ~2 kW/cm² to photoswitch most of the labeled AF647 molecules into a dark state, while a weak (0–1 W/cm²) 405-nm laser switched a small amount of the dark-state AF647 molecules back to the emitting state to allow single-molecule detection. The resultant single-molecule images were recorded using an Andor iXon Ultra 897 EM-CCD at 110 frames per second. 3D-STORM was achieved by inserting a cylindrical lens into the imaging path to introduce astigmatism to the single-molecule images[18]. For most data, the built-in Nikon Perfect Focus System was used to lock the sample focus during data acquisition. For data acquired without using the focal lock, the real-time single-molecule images were monitored so that focal drifts were visually detected and manually corrected during data acquisition.

## Reporting summary

Further information on research design is available in the Nature Portfolio Reporting Summary linked to this article.

## Data availability

Simulated and experimental datasets of this work are made available under Creative Commons Attribution Non Commercial 4.0 International (CC BY-NC 4.0) at: https://doi.org/10.5281/zenodo.16513620.

## Code availability

Codes used in this work are made available under Creative Commons Attribution Non Commercial 4.0 International License (CC BY-NC 4.0) at: https://doi.org/10.5281/zenodo.16513620.

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

## Acknowledgements

L.P. is supported by the National Key Research and Development Program of China (2022YFC3400600), National Natural Science Foundation of China (32227802, 12174208), Guangdong Major Project of Basic and Applied Basic Research (2020B0301030009), Fundamental Research Funds for the Central Universities (63241509), Tianjin Natural Science Foundation (23JCYBJC01250), and the 111 Project (B23045). K.X. acknowledges support from the Packard Fellowships for Science and Engineering and the Heising-Simons Faculty Fellows Award.

## Author contributions

L.P. conceived the research. L.P., K.X., and M.H. designed the theoretical model and experiments. M.H. and J. Y. performed the experiments. M.H., M.Y., F.H., R.Z., and Y.P. analyzed the data. W.L., M.C., and J.X. contributed to the interpretation of the results. K.X. and L.P. wrote the manuscript. L.P., K.X., and J.X. supervised the work.

## Competing interests

The authors declare no competing interests.
