## [Transparent Peer Review file · Nature Communications]

Fast and robust drift correction for single-molecule localization microscopy

Corresponding Author: Professor Ke Xu

A version of this paper was originally rejected for publication by Nature Communications, however that decision was reconsidered after appeal by the authors.

Version 0:

Reviewer comments:

Reviewer #1

(Remarks to the Author)

CONTEXT:

This paper reports an algorithm (NP-Cloud) to determine the drift correction needed for data produced by 2D single-molecule localization microscopy (SMLM) by using the localizations directly. It does this by considering the nearest neighbor distances of the localizations found in a "segment" (collection) of frames as compared to the previous segment using a cutoff criterion if a distance gets too large. The centroid of these relative spatial shifts are used to determine the drift correction. The correction is not always complete, so an iterative algorithm was developed. In a third procedure derived from ideas in Redundant Cross Correlation (RCC), the entire (partially) drift-corrected dataset is subsampled after the first round of NP-Cloud, and the basic NP-Cloud algorithm is applied a second time (resampled reference NP-Cloud or RR NP-Cloud). The time needed to compute drift correction is fast, at least compared to RCC, and the algorithms are simple and easy to understand.

We note that NP (Nearest Paired cloud) as described in the paper is more commonly known as nearest neighbor (NN) --- see, for example, the knnsearch function in MATLAB.

The major problem with this paper is that several recent articles have been missed or not given their due credit (see bib entries at the bottom), omitting the context of idea development as well as newer codes to compare against. In Han2015, a molecular constraint function (MCF) is applied between the points in two images, where the MCF is an exponentially decreasing function of the distances. Collections ("segments") of frames were used here for comparison.

[Fazel2019] used clustered localizations and their uncertainties in a Bayesian statistical framework to determine (possibly drifted) positions of emitters. [Wester2021], in a post-processing drift correction algorithm, grouped localizations in datasets ("segments") and performed both intra-dataset and inter-dataset drift correction using nearest neighbor distances limited by a cutoff criterion somewhat similar to the current paper. Also, the results of this algorithm were compared to RCC on several data collections, simulated and real, running much faster than RCC and producing sharper results in general. [SchodtWester2023] implements a version of the above in an SMLM analysis

package.

[Ma2024], like RR NP-Cloud, uses a two stage process to compute drift, the first to do the basic correction and the second to refine the results. Another recent citation missing is [Fazekas2021].

The way the paper is written, it sounds like many of the ideas therein are completely new. It is important to put them in the context of previous works, especially those with ideas that are highly related to the current article. It also important to compare NP-Cloud with the above algorithms, especially [Wester2021], which is available online in MATLAB (see [SchodtWester2023] and included code: "run_compare.m").

It is not clear when NP-Cloud or iterative NP-Cloud is used subsequent to the latter's introduction in Fig. 1.

ORGANIZATION:

"A user-friendly open-source code" (at the end of the Introduction) should mention that the code is written in MATLAB, instead of digging in the Methods for this information.

A number of times text and figures reference acronyms or concepts not previously defined, but show up later. For example, the introduction of Fig. 2b, etc. comes before RCC and RR NP-Cloud are first mentioned in the text. This makes the paper somewhat confusing on the first read-through.

METHODS:

Section: NP-Cloud implementation for extracting spatial shifts.

- o What is the termination criterion for iterative NP-Cloud?
- o How well the does the iterative approach mitigate background asymmetry?
- o Are there any limitations/errors introduced in this process considering the incremental nature of the shift adjustment process?

Section: NP-Cloud implementation for drift correction of SMLM data.

- o What potential limitations might arise from this method in terms of cumulative errors?

Section: RR-NP Cloud implementation.

- o What are the implications of subsampling and increasing the reference density on the resolution and reliability of the final corrected dataset?

TESTING:

It is important to have some quantitative measures of image quality/sharpness, such as the Fourier Ring Correlation (FRC), otherwise, the results obtained are subjective, especially when working with real data. Some of the papers referenced, including the Ma2024 paper, use FRC for some of their comparisons.

How important is it to have SMLM "segments" of just a few frames (like 15) rather than 50 or 100? A big deal is made of this. Obviously, too large of a segment loses fine detail, but too small has few localizations and is prone to spurious detail.

3D:

No mention is made of 3D data, an important measurement in recent SMLM.

CODE/DATA:

A README file is lacking. Run_NPC_RRNPC.m is designed to run from Windows; we

generalized it to run under Windows/Linux by changing lines like

```
load '..\SimulatedData\SimulatedSMLM_HighDensity100.mat';
```

to

```
load(fullfile('..', 'SimulatedData', 'SimulatedSMLM_HighDensity100.mat'));
```

etc.

The figure produced is missing axis labels. It is also difficult to see the differences where magenta and dashed red lines overlap (green and dashed blue is clearer). Moreover, running an example file overrides the saved results produced by the previous file because the name of the file is not part of the name of the saved results.

The code is useful with the changes made above, but it would be nice to be made prettier as well as converted into a function that can be simply called by the user with well documented inputs and outputs.

ENGLISH:

I would suggest that the authors employ a native English speaker to read through the text as there are places where the words, used improperly or are missing, are distracting. For example,

2nd paragraph on page 2: trial-and-errors

middle of page 4: clusters of single-molecule localizations _were_ distributed

starting bottom line of page 5: could be similarly implemented in a redundant

approach and was ~500-fold fold faster than RCC [hard to read and follow!]

3rd line of page 6: NP-Could [typo]

bottom of page 7: a second-pass NP-Could analysis, [typo]

2nd paragraph on page 8: single-molecule localizations that _were_ scattered with standard deviations of

5th line on page 12: 1,600 localizations in the red channel across the entire field were each searched for its [-> their] nearest paired localization

Figure S3 caption: to minimize -> that minimizes

Description of Supplementary Codes and Data, 5th line: See noted in comments.

Commas also need to be added/removed in various places.

% Maximizes the molecular constraint field (MCF) cost function, where the MCF % is a function of the distances between the points in two images. For

% example,

%

% $MCF(L) = \sum_i \sum_j \frac{1}{2} \pi \sigma^2$

% $e^{-\frac{1}{2} (L(p_j) - p_i)^2 / \sigma^2}$

%

% where σ is a constant, $L(\cdot)$ is a transformation function, and
% $p_i \in \phi_A$ and $p_j \in \phi_B$ are position vectors defining the points in
% the datasets ϕ_A and ϕ_B , respectively. Here, ϕ_A is assumed fixed
% and ϕ_B movable. The MCF takes into account the sparseness of points
% around each position p_i , and produces near zero contribution when the
% distances become large. σ is taken to be at least twice the standard
% deviation of the localization precision. Typical usage involves combining
% collections of sequential frames into datasets over correspondingly longer
% time intervals to which the above cost function is applied.

@ARTICLE{Han2015,

AUTHOR = "Renmin Han and Liansan Wang and Fan Xu and Yongdeng Zhang and Mingshu Zhang and Zhiyong Liu and Fei Ren and Fa Zhang",

TITLE = {Drift-correction for single-molecule imaging by molecular constraint field, a distance minimum metric},

JOURNAL = "BMC Biophysics",

VOLUME = 8,

NUMBER = 1,

YEAR = 2015,

PAGES = "1--14",

DOI = "10.1186/s13628-014-0015-1"

}

[Fazel2019]

Mohamadreza Fazel, Michael~J. Wester, Hanieh Mazloom-Farsibaf, Marjolein B.~M. Meddens, Alexandra~S. Eklund, Thomas Schlichthaerle, Florian Schueder, Ralf Jungmann and Keith~A. Lidke, ``Bayesian Multiple Emitter Fitting using Reversible Jump Markov Chain Monte Carlo'', {sl Scientific Reports}, Volume~9, Article~13791, September~24, 2019, 1--10 (DOI: 10.1038/s41598-019-50232-x).

[Wester2021]

Michael~J. Wester, David~J. Schodt, Hanieh Mazloom-Farsibaf, Mohamadreza Fazel, Sandeep Pallikkuth and Keith~A. Lidke, ``Robust, fiducial-free drift correction for super-resolution imaging'', {sl Scientific Reports}, Volume~11, Article~23672, December~8, 2021, 1--14, {sf <https://www.nature.com/articles/s41598-021-02850-7>} (DOI: 10.1038/s41598-021-02850-7).

[Fazekas2021]

A mean shift algorithm for drift correction in localization microscopy
Frank J. Fazekas, Thomas R. Shaw, Sumin Kim, Ryan A. Bogucki, Sarah L. Veatch
Biophysical Reports, Volume 1, Issue 1, 100008, September 08, 2021

[SchodtWester2023]

David~J. Schodt*, Michael~J. Wester*, Mohamadreza Fazel, Sajjad Khan, Hanieh Mazloom-Farsibaf, Sandeep Pallikkuth, Marjolein B.~M. Meddens, Farzin Farzam, Eric~A. Burns, William~K. Kanagy, Derek~A. Rinaldi, Elton Jhamba, Sheng Liu, Peter~K. Relich, Mark~J. Olah, Stanly~L. Steinberg and Keith~A. Lidke (* = co-1st author), "SMITE: Single Molecule Imaging Toolbox Extraordinaire (MATLAB)", {sl Journal of Open Source Software}, Volume~8, Number~90, 2023, p. 5563, {sf <https://joss.theoj.org/papers/10.21105/joss.05563>}, (DOI: 10.21105/joss.05563).

[Ma2024]

Ma, H., Chen, M., Nguyen, P. & Liu, Y. Toward drift-free high-throughput nanoscopy through adaptive intersection maximization. Science Advances 10, eadm7765 (2024).

(Remarks on code availability)

CODE/DATA:

A README file is lacking. Run_NPC_RRNP.m is designed to run from Windows; we generalized it to run under Windows/Linux by changing lines like
load '..\SimulatedData\SimulatedSMLM_HighDensity100.mat';
to
load(fullfile('..', 'SimulatedData', 'SimulatedSMLM_HighDensity100.mat'));
etc.

The figure produced is missing axis labels. It is also difficult to see the differences where magenta and dashed red lines overlap (green and dashed blue is clearer). Moreover, running an example file overrides the saved results produced by the previous file because the name of the file is not part of the name of the saved results.

The code is useful with the changes made above, but it would be nice to be made prettier as well as converted into a function that can be simply called by the user with well documented inputs and outputs.

We were able to reproduce the drift correction curves for various datasets supplied with the paper (but not other results such as figures of the corrected data, at least with the software supplied).

Reviewer #2

(Remarks to the Author)

(Remarks on code availability)

Reviewer #3

(Remarks to the Author)

In the present manuscript, the authors have implemented a method for correcting the physical drift that occurs during SMLM data acquisition. It is based on estimating the vectorial displacements for nearest paired particle positions between two consecutive segments (i.e., set of aggregated consecutive frames) of SMLM dataset, whereas the standard methods are based on searching the maximum correlation between the two reconstructed SMLM images from two sets of consecutive frames).

The authors first validate the NP-Cloud approach on simulated data. The data were simulated for two different densities of particles/ μm^2 on which a drift is applied in x and y directions. Knowing the ground truths of their SMLM datasets, they compare the results of NP-Cloud to others, to conclude to the robustness and fast speed of the NP-Cloud analyses. Next, the authors apply their drift correction on experimental SMLM data.

Overall, I found the work potentially interesting. I fully agree that a robust drift correction method is important to ensure the validity of SMLM analyses. In this context, current drift correction methods are part of standard post-acquisition procedures, which does not pose a major problem in terms of time savings. Therefore, as long as a drift estimation combined with corrective approaches is not implemented in real-time during the acquisition process, the NP-Cloud method would have only a limited impact for the scientific community using SMLM observations.

Moreover, in its current form, I feel that substantial clarifications and modifications need to be made to consolidate the conclusion of this work.

Generally speaking, without a description of the NP-Cloud workflow as usually provided to illustrate the principle of an algorithm, it is difficult to understand what exactly is being calculated without analyzing the code in detail. Similarly, the assumptions about NP-Cloud required to be valid/applicable are not indicated. Moreover, as NP-Cloud requires the adjustment of parameters, it would be important to mention them explicitly, as well as to give the numerical minima/maxima of these parameters.

The conditions of the simulations are rather too optimistic compared with the data usually recorded by SMLM. For example, obtaining a standard deviation of 10 nm for sparse particles corresponds to an SNR of over 30 dB (usually, the average SNR value is around 25 dB). In addition, the density of particles/frame/ μm^2 is very low (≈ 0.02 particles/frame/ μm^2), a condition at least 10 times lower than those usually encountered in SMLM acquisition; since even weakly fluorescent particles distort the accuracy of the localization of relatively distant particles (by biasing the fit of the energy distribution in the pixels), this would also distort the drift correction. However, the overall drift values applied in the x and y axes are quite high, i.e., of the same order as the accuracy; in fact, the drift must necessarily be lower than this value, otherwise the SMLM observations will not be validly informative. So, given that all these factors (SNR, density/frame/ μm^2 , drift) would have a significant impact on the method, it would be important to assess its robustness on more realistic ground truth data.

Finally, from my point of view, the major added-value of NP-Cloud algorithm lies not in its computational speed, but rather in the fact that the method does not priori require a reference, i.e., a structure (fibers or organelles, or beads) to be of more general interest. As such, this work would benefit to elaborate more specifically this aspect with simulated and experimental data.

(Remarks on code availability)

I have barely been able to examine the code - I couldn't get into the details.

Reviewer #4

(Remarks to the Author)

In this paper, Hou, et al. present a drift correction algorithm for single-molecule localization microscopy data based on the nearest paired cloud (NP-Cloud). They test the method using both simulated and biological data. However, I do not recommend the paper for publication in Nature Communications due to its lack of novelty and impact. There are already numerous drift correction methods available, including one that is very similar to the proposed approach (see comment #1). Further, I am not convinced that the method described in this paper provides a significant improvement over the existing drift correction methods. Below are my main concerns:

1) The authors failed to cite the following drift correction method that uses a very similar approach (nearest neighbor distance) to correct for drift:

MJ Wester et al, Scientific Reports 11(1), 1-14, 2021.

Authors must mention how NP-Cloud is distinguished from the method described in this paper as well as providing comparison of their results with this method.

2) While the NP-Cloud algorithm is capable of correcting for 3D drift, it is not mentioned in the main text. I recommend discussing the 3D drift correction capability of the NP-Cloud algorithm in the abstract, introduction and conclusions. Further, I also suggest moving Fig. S7 to the main text and testing the algorithm on at least one 3D synthetic data.

3) SI figures are not discussed at all. These figures need to be referred to and discussed in the Results section in the main text.

Thanks

(Remarks on code availability)

While the code seems to work they way described by the authors, they failed to annotate the code. It does not have any description of the inputs and outputs and what is the purpose of each function.

Version 1:

Reviewer comments:

Reviewer #1

(Remarks to the Author)

Nature Communications manuscript NCOMMS-24-58384-T
Nearest paired cloud (NP-Cloud): Fast robust drift correction for single-molecule localization microscopy

Mengdi Hou, Jianyu Yang, Fen Hu, Rongge Zhao, Yuhang Pan, Mingjie Yang, Wan Li, Mingxin Chen, Jingjun Xu, Ke Xu, Leiting Pan

This paper revision is certainly much improved over the original manuscript. Many of our concerns have been well addressed. There are still a few outstanding issues as described below.

cumulative errors

In your response, you say there are no cumulative errors in the NP-Cloud algorithm, however, in the last lines of page 10 you say: Summing the shifts over the iterative steps yielded a cumulative shift ..., which seems to contradict your response in this instance, that is, there are cumulative iterative calculations. We don't think the error is great, but there is some.

clusters

We understand that clusters of localizations are produced in the simulations. In Figure 3c, SMLM cluster FWHM are computed for different algorithms and segment sizes. It is not clear to us what exactly is being shown and how the clusters are determined in the analysis. This needs to be made clearer as the mention here is confusing.

3D Data (Z)

You talk about focal locks in commercial microscopes, but, of course, you are excluding those microscopes that do not use focal locks, or make use of z-stacks, or other 3D setups employed in custom-built microscopes. It seems likely that you could expand your algorithm to 3D; 3D is the up and coming trend in microscope design. For sure, you should mention in the abstract that the algorithm is 2D if you are not willing to generalize it, which we understand would require substantial additional work.

typos

Page 5, line 6: ,, -> ,
Page 10, middle: SMLM acquisitions. -> SMLM acquisition.
SI Figure 3a and b: "resundant-NPC" in the two bottom left plots in their legends

software

Much improved. The output directory name is: DirftCurveOutput, which has an obvious typo. It would be (as suggested in the original review) nice if the output directory name and/or output filenames somehow identify their data source, so running a different dataset does not overwrite an earlier result from a different source (different path/file).

(Remarks on code availability)

Much improved. The output directory name is: DirftCurveOutput, which has an obvious typo. It would be (as suggested in the original review) nice if the output directory name and/or output filenames somehow identify their data source, so running a different dataset does not overwrite an earlier result from a different source (different path/file).

Documentation is much better and the software is quite usable.

Reviewer #3

(Remarks to the Author)

In this new version of their manuscript, the authors have clarified the various points that I had initially raised.

(Remarks on code availability)

I ran the code on simulated and experimental data, which convinced me of the added value of the method.

Reviewer #4

(Remarks to the Author)

This manuscript is improved and most of my concerns have been addressed. I specially appreciate the addition of SI Fig. 8-10 to compare the proposed method to the previous approaches. However, I still believe that the proposed method does not show significant improvement over existing methods and another journal, like Scientific Reports, might be a more suitable place for this paper. It is up to the editors to make this decision.

(Remarks on code availability)

Version 2:

Reviewer comments:

Reviewer #1

(Remarks to the Author)

Nature Communications manuscript NCOMMS-24-58384-T

Nearest paired cloud (NP-Cloud): Fast robust drift correction for single-molecule localization microscopy

Mengdi Hou, Jianyu Yang, Fen Hu, Rongge Zhao, Yuhang Pan, Mingjie Yang, Wan Li, Mingxin Chen, Jingjun Xu, Ke Xu, Leiting Pan

The manuscript looks great and ready to publish after fixing a few small typos:

page 4, next to the last paragraph:

also, page 11, new material:

`_is_` reported on the initial displacement between the two channels,

page 9, end of 1st full paragraph

ground-truth drift trajectory, with the in-plan`_e_`xy shift being identical

(Remarks on code availability)

All my earlier comments have been addressed.

Reviewer #1 (Remarks to the Author):

CONTEXT:

This paper reports an algorithm (NP-Cloud) to determine the drift correction needed for data produced by 2D single-molecule localization microscopy (SMLM) by using the localizations directly. It does this by considering the nearest neighbor distances of the localizations found in a "segment" (collection) of frames as compared to the previous segment using a cutoff criterion if a distance gets too large. The centroid of these relative spatial shifts are used to determine the drift correction. The correction is not always complete, so an iterative algorithm was developed. In a third procedure derived from ideas in Redundant Cross Correlation (RCC), the entire (partially) drift-corrected dataset is subsampled after the first round of NP-Cloud, and the basic NP-Cloud algorithm is applied a second time (resampled reference NP-Cloud or RR NP-Cloud). The time needed to compute drift correction is fast, at least compared to RCC, and the algorithms are simple and easy to understand.

Response: We thank Reviewers #1 and #2 for their excellent summary of our work and for noting that our approach is “fast” and “simple and easy to understand”.

We note that NP (Nearest Paired cloud) as described in the paper is more commonly known as nearest neighbor (NN) --- see, for example, the knnsearch function in MATLAB.

Response: Thank you for this discussion. We have added a mention of “nearest neighbor” to text. We prefer the term “nearest pair” as we are trying to pair each molecule in a segment with those in the reference segment. Our starting point was the pairing of localizations between different frames, like what we did in our recent development of SMdM (Refs 19-21). Our initial test of the knnsearch function in MATLAB indicated it to be less effective than our current NP-Cloud code.

The major problem with this paper is that several recent articles have been missed or not given their due credit (see bib entries at the bottom), omitting the context of idea development as well as newer codes to compare against. In Han2015, a molecular constraint function(MCF) is applied between the points in two images, where the MCF is an exponentially decreasing function of the distances. Collections ("segments") of frames were used here for comparison.

Response: Thank you for this discussion. We have previously focused on discussion and comparison with the traditional (“standard”) drift-correction approaches developed for STORM by Prof. Xiaowei Zhuang and Prof. Bo Huang, namely DCC and RCC. In this revision, we have substantially expanded our discussion on comparison with other emerging drift-correction approaches. See our revised introduction, new discussion on Page 8, and new **Supplementary Figs. 8-10**. This new analysis shows that RR NP-Cloud is the most robust method, uniquely providing fast, optimal drift correction. See also response to the questions below.

[Fazel2019] used clustered localizations and their uncertainties in a Bayesian statistical framework to determine (possibly drifted) positions of emitters. [Wester2021], in a post-processing drift correction algorithm, grouped localizations in datasets ("segments") and performed both intra-dataset and inter-dataset drift correction using nearest neighbor distances limited by a cutoff criterion somewhat similar to the current paper. Also, the results of this algorithm were compared to RCC on several data collections, simulated and real, running much faster than RCC and producing sharper results in general. [SchodtWester2023] implements a version of the above in an SMLM analysis package.

Response: Thank you for this discussion. We have added discussion on these approaches and included a side-by-side comparison of performance. We thank the reviewer for providing us with the

“run_compare.m” code for [Wester2021] and its recent MATLAB implementation [SchodtWester2023]. We have run it (without any changes to the set parameters in this file) on our data for a side-by-side comparison. We thus found that although [Wester2021] (driftCorrectKNN) is reasonably fast, the residual errors are substantially larger than both NPC and RR-NPC. We have added these results and discussion to Page 8 and new **Supplementary Figs 8-10**.

[Ma2024], like RR NP-Cloud, uses a two stage process to compute drift, the first to do the basic correction and the second to refine the results. Another recent citation missing is [Fazekas2021].

Response: We clarify that we actually cited both these two studies in our previous manuscript as References 14 and 15, although we did not discuss them in detail. In this revision, we have expanded our discussions and included side-by-side comparisons of performance. From this comparison, we note that since [Ma2024] (adaptive intersection maximization; AIM) is still based on the initial gridding/pixelization of the single-molecule localizations to correlate the gridded/pixelated SMLM images, it has the same issue as DCC/RCC in failing at small segment sizes, under which conditions insufficient localizations are counted in each gridded bin. While [Fazekas2021] (mean shift) produces good drift correction results that are comparable to RR-NPC, it is excessively slow due to its full redundancy approach like RCC. It is $\sim 100x$ slower than RR-NPC for large segment sizes, but even much slower for small segment sizes. For example, for our simulated SMLM data of 66,000 frames, [Fazekas2021] (mean shift) took 6×10^5 s (172 hours or 7 days) at a segment size of 50 frames per segment, and was unable to complete over 15 days for the segment size of 30 frames per segment. In contrast, RR-NPC was completed in 10 s for both cases with good results. We have added these results and discussion to Page 8 and new **Supplementary Figs 8-10**.

The way the paper is written, it sounds like many of the ideas therein are completely new. It is important to put them in the context of previous works, especially those with ideas that are highly related to the current article. It also important to compare NP-Cloud with the above algorithms, especially [Wester2021], which is available online in MATLAB (see [SchodtWester2023] and included code: "run_compare.m").

Response: Thank you for this discussion and for kindly providing us with the code to test [Wester2021]/[SchodtWester2023]. As discussed above, we have used it to generate a side-by-side comparison. As mentioned above, we have also substantially expanded our discussion on comparison with other emerging drift-correction approaches.

It is not clear when NP-Cloud or iterative NP-Cloud is used subsequent to the latter's introduction in Fig. 1.

Response: The iterative algorithm is an integral part of NP-Cloud to determine the cloud center, so it is always used in all NP-Cloud calculations. We have clarified this in the revision.

ORGANIZATION:

"A user-friendly open-source code" (at the end of the Introduction) should mention that the code is written in MATLAB, instead of digging in the Methods for this information.

Response: Thank you for this discussion. We have revised this text to say “An open-source MATLAB code”.

A number of times text and figures reference acronyms or concepts not previously defined, but show up later. For example, the introduction of Fig. 2b, etc. comes before RCC and RR NP-Cloud are first mentioned in the text. This makes the paper somewhat confusing on the first read-through.

Response: Thank you for this discussion. We have now spelled out the acronyms in the caption.

METHODS:

Section: NP-Cloud implementation for extracting spatial shifts.

o What is the termination criterion for iterative NP-Cloud?

Response: The iteration repeated until convergence. As shown in **Fig. 1g**, as the shifting continues to move the cloud center closer and closer to the origin, the calculated mean position of the cloud also moves closer and closer to the origin. The iteration was terminated when the calculated mean position of the cloud after the shift was no longer closer to the origin when compared to the previous iteration, signaling no further possible improvement with shifting. We have clarified this point in the revised text.

o How well the does the iterative approach mitigate background asymmetry?

Response: As shown in **Fig. 1d-g**, the iterative process fully removed the background asymmetry. **Fig. 1g** shows that we recovered the true drift value in this process. **Fig. 1f** shows that after the iterations, NP-Cloud appeared near-identical to that without shift (**Fig. 1c**), so background asymmetry no longer exists. We have revised our text to mention this.

o Are there any limitations/errors introduced in this process considering the incremental nature of the shift adjustment process?

Response: We did not see issues with the process. The approach works quite well, as shown in **Fig. 1d-g** and also the vast datasets we tested in the drift correction of both simulated and experimental data.

Section: NP-Cloud implementation for drift correction of SMLM data.

o What potential limitations might arise from this method in terms of cumulative errors?

Response: Our approach does not incur “cumulative errors”. Note that instead of comparing the relative drifts between consecutive segments, in which case errors may accumulate, we compare each segment with the same reference (the first segment in NP-Cloud or expanded sampling in RR-NP Cloud). We just pre-move the positions using the calculated drift of the previous segment to minimize the remaining drift to make the search more efficient. Thus, we showed with vast simulated and experimental data excellent drift correction, including those with many frames and large total drifts. We have added a discussion of this to the text.

Section: RR-NP Cloud implementation.

o What are the implications of subsampling and increasing the reference density on the resolution and reliability of the final corrected dataset?

Response: As discussed, the limitation of single-referenced DCC and NP-Cloud is that later segments are compared to only the initial segment. So, if certain structures are not sampled (or under-sampled) in the first segment, which is likely given the stochastic nature of SMLM, they are missed in the drift-correction calculations. Our resampled-reference NP-Cloud approach overcomes this limitation by

evenly expanding the reference size across the full dataset, and we showed with both simulated and experimental data that this approach further improved drift correction without the excessive redundancy of RCC. We have improved our text on related discussions.

TESTING:

It is important to have some quantitative measures of image quality/sharpness, such as the Fourier Ring Correlation (FRC), otherwise, the results obtained are subjective, especially when working with real data. Some of the papers referenced, including the Ma2024 paper, use FRC for some of their comparisons.

Response: Thank you for this discussion. For simulated data, as the ground truth is known, absolute residuals can be obtained directly. We thus focused on using these values to evaluate the effectiveness of drift correction. For experimental data, the drift-correction effects are apparent from the corrected images themselves. For quantification, we started with the data of tropomodulin in erythrocytes: its pattern is isolated clusters, so we can readily use the SMLM cluster size to show how well drift correction worked. For the very dense structure of spectrin in erythrocytes, we used beads to measure the actual drift in the sample while applying different drift-correction methods to data excluding the bead positions, and so we were able to compare the drift-correction performance with the ground truth provided by the beads. FRC is certainly helpful for samples without beads as the ground truth, but is also convolved by the sample structure patterns. In this revision, we have applied FRC to the experimental data of microtubules, which again confirmed superior results by RR-NPC (revised **Supplementary Fig. 5**).

How important is it to have SMLM "segments" of just a few frames (like 15) rather than 50 or 100? A big deal is made of this. Obviously, too large of a segment loses fine detail, but too small has few localizations and is prone to spurious detail.

Response: Thank you for this discussion. (RR)-NP-Cloud is characterized by excellent robustness for varied segment sizes. Especially for small segment sizes, when the count of localizations is low, RR-NP Cloud uniquely provides robust drift correction when other approaches fail, due to its efficient use of the super-localized positions of all molecules. The workable range of segment size depends on the sample, *e.g.*, how the localizations are distributed.

For the simulated dataset of 66,000 frames with features of 100 clusters/ μm^2 , in **Fig. 2f** and **Supplementary Fig. 2** we show DCC and RCC do not work properly at segment sizes of 50 frames or less. In this revision, we further show that emerging methods, driftCorrectKNN, mean shift, and AIM, all failed at 30 frames per segment (**Supplementary Figs. 8** and **10**). NP-Cloud worked well for all segment sizes (**Fig. 2f** and **Supplementary Fig. 2**).

For experimental data, we show in **Fig. 3** notable issues of DCC and RCC for segment sizes of 100 frames or less. In **Supplementary Fig. 6**, we further show an interesting result in which at 200 frames per segment, DCC initially worked but failed for later parts of the data, when the count of localizations in each frame dropped due to photobleaching, a common scenario in SMLM. Thus, even in the same experiment with a fixed segment size, reduced counts could deteriorate drift correction for later parts of the data. RR-NP-Cloud robustly handled the data without issues, and worked well even for the much smaller segment size of 50 frames/segment, under which condition DCC completely failed.

We have incorporated the above discussion in our revision.

3D:

No mention is made of 3D data, an important measurement in recent SMLM.

Response: Thank you for this discussion. For 3D drift correction, even though NP-Cloud could be conceivably extended to also correct drift in depth (z), there is limited practical need, since modern commercial microscopes are often equipped with focal locks precise to ~ 10 nm, better than the typical z resolution of SMLM (50 nm FWHM). Importantly, since the imaging depth of 3D-SMLM is very limited (± 400 nm of the focal center), if significant z drift ever occurs during imaging, then the same structures may not be recaptured with the shifted focus, and drift correction cannot be achieved through algorithms. Consequently, focal locks are typically used in SMLM. The 3D-STORM data in Fig. S7 used NP-Cloud to drift correct xy while leaving the z values unchanged. As can be seen, the final results were excellent, correctly showing the hollow structure of mitochondria. This result is representative of typical 3D-STORM, in which z has minimal drift under focal lock, whereas xy drift deteriorates the image and needs correction. We have discussed the above points in our revised manuscript.

CODE/DATA:

A README file is lacking. Run_NPC_RRNPC.m is designed to run from Windows; we generalized it to run under Windows/Linux by changing lines like
load '..\SimulatedData\SimulatedSMLM_HighDensity100.mat';
to
load(fullfile('..', 'SimulatedData', 'SimulatedSMLM_HighDensity100.mat'));
etc.

The figure produced is missing axis labels. It is also difficult to see the differences where magenta and dashed red lines overlap (green and dashed blue is clearer). Moreover, running an example file overrides the saved results produced by the previous file because the name of the file is not part of the name of the saved results.

Response: Thank you for helping us generalize the codes. We have added axis labels and improved the color contrasts in the plots. We provide the raw codes and data as open-source, so everyone is welcome to use and modify the codes and present/save the results with their personal preferences.

The code is useful with the changes made above, but it would be nice to be made prettier as well as converted into a function that can be simply called by the user with well documented inputs and outputs.

Response: Thank you for testing and validating our codes. Our current call function “NPC_RRNPC_CallFunction.m” is aimed to provide this service, so that raw single-molecule localizations in each frame are the inputs, and the drift correction curves and corrected localizations are the outputs. In this revision we have added annotations to better explain the inputs and outputs.

ENGLISH: I would suggest that the authors employ a native English speaker to read through the text as there are places where the words, used improperly or are missing, are distracting. For example,

2nd paragraph on page 2: trial-and-errors

middle of page 4: clusters of single-molecule localizations were distributed

starting bottom line of page 5: could be similarly implemented in a redundant approach and was ~ 500 -fold fold faster than RCC [hard to read and follow!]

3rd line of page 6: NP-Could [typo]

bottom of page 7: a second-pass NP-Could analysis, [typo]

2nd paragraph on page 8: single-molecule localizations that were scattered with standard deviations of

5th line on page 12: 1,600 localizations in the red channel across the entire

field were each searched for its [-> their] nearest paired localization

Figure S3 caption: to minimize -> that minimizes

Description of Supplementary Codes and Data, 5th line: See noted in comments.

Commas also need to be added/removed in various places.

Response: Thank you for the suggestion. We have addressed the specific points listed above and also asked a native English speaker to help polish our text.

Reviewer #1 (Remarks on code availability):

This section duplicates the “CODE/DATA:” section above.

Reviewer #2 (Remarks to the Author):

Reviewer #3 (Remarks to the Author):

In the present manuscript, the authors have implemented a method for correcting the physical drift that occurs during SMLM data acquisition. It is based on estimating the vectorial displacements for nearest paired particle positions between two consecutive segments (i.e., set of aggregated consecutive frames) of SMLM dataset, whereas the standard methods are based on searching the maximum correlation between the two reconstructed SMLM images from two sets of consecutive frames).

The authors first validate the NP-Cloud approach on simulated data. The data were simulated for two different densities of particles/ μm^2 on which a drift is applied in x and y directions. Knowing the ground truths of their SMLM datasets, they compare the results of NP-Cloud to others, to conclude to the robustness and fast speed of the NP-Cloud analyses. Next, the authors apply their drift correction on experimental SMLM data.

Overall, I found the work potentially interesting. I fully agree that a robust drift correction method is important to ensure the validity of SMLM analyses. In this context, current drift correction methods are part of standard post-acquisition procedures, which does not pose a major problem in terms of time savings. Therefore, as long as a drift estimation combined with corrective approaches is not implemented in real-time during the acquisition process, the NP-Cloud method would have only a limited impact for the scientific community using SMLM observations.

Response: We thank the reviewer for his/her kind summary of our work and for noting its potential significance.

Moreover, in its current form, I feel that substantial clarifications and modifications need to be made to consolidate the conclusion of this work.

Generally speaking, without a description of the NP-Cloud workflow as usually provided to illustrate the principle of an algorithm, it is difficult to understand what exactly is being calculated without analyzing the code in detail. Similarly, the assumptions about NP-Cloud required to be valid/applicable are not

indicated. Moreover, as NP-Cloud requires the adjustment of parameters, it would be important to mention them explicitly, as well as to give the numerical minima/maxima of these parameters.

Response: Thank you for the discussion. The central concept of NP-Cloud is described in Fig 1, where we explained the approach and validated how it can be used to extract nanoscale spatial shifts between two sets of single-molecule localizations, *e.g.*, a segment of SMLM data versus the reference segment in drift correction. We have revised our text to better explain our approach, and also improved annotations in our code.

We note that one major advantage of NP-Cloud is its robustness and carefree nature, with no need to guess parameters and run trial runs. In all the results we showed in the paper on diverse simulated and experimental data, we did not need to adjust any parameters. We just compared the results at different segment sizes, *i.e.*, the number of frames in each segment for drift correction, with other methods. And even for that single parameter, we showed that RR-NP-Cloud was highly robust for typical values of 50-200 for vast simulated and experimental data, even as other approaches failed, and generally provided better final drift-correction results. We have improved the related discussion in the text., *e.g.*, in Conclusion: “It is further worth noting that except for the segment size, which may be optimized based on the single-molecule localization density, NP-Cloud and RR-NP require no adjustment of parameters when running on diverse datasets.” In the code we provide, we also now include only a single “Adjusted parameter”, the segment size.

The conditions of the simulations are rather too optimistic compared with the data usually recorded by SMLM. For example, obtaining a standard deviation of 10 nm for sparse particles corresponds to an SNR of over 30 dB (usually, the average SNR value is around 25 dB). In addition, the density of particles/frame/ μm^2 is very low (≈ 0.02 particles/frame/ μm^2), a condition at least 10 times lower than those usually encountered in SMLM acquisition; since even weakly fluorescent particles distort the accuracy of the localization of relatively distant particles (by biasing the fit of the energy distribution in the pixels), this would also distort the drift correction. However, the overall drift values applied in the x and y axes are quite high, *i.e.*, of the same order as the accuracy; in fact, the drift must necessarily be lower than this value, otherwise the SMLM observations will not be validly informative. So, given that all these factors (SNR, density/frame/ μm^2 , drift) would have a significant impact on the method, it would be important to assess its robustness on more realistic ground truth data.

Response: Thank you for this discussion. Our simulation data were based on our typical running conditions of STORM: As frequently demonstrated by us and originally by Xiaowei Zhuang Lab, STORM routinely obtains localization precisions of 10 nm, as determined by comparing the repeated localizations of the same single molecule, which is effectively what we simulated. We have added references to this discussion in the text.

Our simulation is based on 30 localizations in each frame, and that value is also typical in our STORM experiments as we record at 110 frames per second. See the experimental data we provide in the supplement: the typical range is 20-50 localizations in each frame. Higher single-molecule densities may be encountered for certain samples or for slower framerates. However, the actual count per frame in the simulation is not a critical factor for our drift-correction results, as we started by combining many frames into segments and then calculated the nearest pairs between segments. Thus, a simulation of 60 localizations per frame at a segment size of 50 frames per segment would be equivalent to a simulation of 30 localizations per frame at a segment size of 100 frames per segment. Higher single-molecule densities may lead to other issues like overlapping single-molecule images, but these are handled with different raw-image fitting algorithms. In this work we focus on the later step of drift correction, and our results on diverse experimental data showed excellent results. We have added related discussion to the text.

In SMLM, the overall drift values are typically larger than the localization uncertainties, and that is why raw SMLM images often appear blurry without drift correction. Meanwhile, drift in each frame and in each segment is certainly smaller than the localization. This fact enabled us to use a small search radius when comparing drift between segments. To be faithful to typical experimental conditions, we actually measured the drift of our microscope using beads, and used the resultant experimental drift values as ground truth for our simulated data. We have improved related discussion in the revision.

Finally, from my point of view, the major added-value of NP-Cloud algorithm lies not in its computational speed, but rather in the fact that the method does not priori require a reference, i.e., a structure (fibers or organelles, or beads) to be of more general interest. As such, this work would benefit to elaborate more specifically this aspect with simulated and experimental data.

Response: Thank you for this discussion. Indeed, NP-Cloud does not need a structure and works robustly for diverse samples, as demonstrated in our supplementary figures. We have expanded our discussion in the revised manuscript.

Reviewer #3 (Remarks on code availability):

I have barely been able to examine the code - I couldn't get into the details.

Reviewer #4 (Remarks to the Author):

In this paper, Hou, et al. present a drift correction algorithm for single-molecule localization microscopy data based on the nearest paired cloud (NP-Cloud). They test the method using both simulated and biological data. However, I do not recommend the paper for publication in Nature Communications due to its lack of novelty and impact. There are already numerous drift correction methods available, including one that is very similar to the proposed approach (see comment #1). Further, I am not convinced that the method described in this paper provides a significant improvement over the existing drift correction methods. Below are my main concerns:

Response: Thank you for this discussion. NP-Cloud provides a conceptually simple, robust, and carefree solution to SMLM drift correction. It requires no second-guessing of “workable” parameters, and is highly robust with better final results over traditional drift-correction methods. It is also >100-fold faster over traditional single-referenced approaches and $>10^4$ faster over cross-referenced redundant approaches. These are demonstrated in our side-by-side comparisons. In this revision we further provide additional comparisons with other emerging approaches. See revised text and new **Supplementary Figs. 8-10**.

1) The authors failed to cite the following drift correction method that uses a very similar approach (nearest neighbor distance) to correct for drift:

MJ Wester et al, Scientific Reports 11(1), 1-14, 2021.

Authors must mention how NP-Cloud is distinguished from the method described in this paper as well as providing comparison of their results with this method.

Response: Thank you very much for this discussion. We have added discussions on this and other emerging approaches and included a side-by-side comparison of performance. For the specific reference of [Wester2021], Reviewer 1 also referred to this work and kindly wrote a code for us to run it using a

recent MATLAB implementation [SchodtWester2023]. We have run it directly on our data for a side-by-side comparison. We thus found that although [Wester2021] (driftCorrectKNN) is reasonably fast, the residual drift-correction errors are substantially larger than both NPC and RR-NPC. Please see our new discussion on Page 8 and new **Supplementary Figs. 8-10**.

2) While the NP-Cloud algorithm is capable of correcting for 3D drift, it is not mentioned in the main text. I recommend discussing the 3D drift correction capability of the NP-Cloud algorithm in the abstract, introduction and conclusions. Further, I also suggest moving Fig. S7 to the main text and testing the algorithm on at least one 3D synthetic data.

Response: Thank you for this discussion. For 3D drift correction, even though NP-Cloud could be conceivably extended to also correct drift in depth (z), there is limited practical need, since modern commercial microscopes are often equipped with focal locks precise to ~ 10 nm, better than the typical z resolution of SMLM (50 nm FWHM). Importantly, since the imaging depth of 3D-SMLM is very limited (± 400 nm of the focal center), if significant z drift really occurs during imaging, then the same structures may not be captured with the shifted focus, and drift correction cannot be achieved through algorithms. Consequently, focal locks are typically used in SMLM. The 3D-STORM data in Fig. S7 used NP-Cloud to drift correct xy while leaving the z values unchanged. As can be seen, the final results were excellent, correctly showing the hollow structure of mitochondria. This result is representative of typical 3D-STORM, in which z has minimal drift under focal lock whereas xy needs drift correction. We have clarified and discussed the above points in our revised manuscript.

3) SI figures are not discussed at all. These figures need to be referred to and discussed in the Results section in the main text.

Response: Thank you for this valuable discussion. We have added additional discussion referring to each SI figure. Among these, we highlight in **Supplementary Fig. 6** an example in which at 200 frames per segment, DCC initially worked but failed for later parts of the data when the count of localizations in each frame reduced due to photobleaching, a common scenario in SMLM. RR-NP Cloud robustly handled the whole dataset, and worked well even for the much smaller segment size of 50 frames/segment, under which condition DCC completely failed.

Thanks

Reviewer #4 (Remarks on code availability):

While the code seems to work they way described by the authors, they failed to annotate the code. It does not have any description of the inputs and outputs and what is the purpose of each function.

Response: Thank you for confirming that our code works. We have added annotations to our code and provided descriptions of the inputs and outputs.

Response to Reviewers' comments

Reviewer #1 (Remarks to the Author):

This paper revision is certainly much improved over the original manuscript. Many of our concerns have been well addressed. There are still a few outstanding issues as described below.

Response: We thank the reviewer for approving our revision. Please see below our new updates and discussion.

cumulative errors

In your response, you say there are no cumulative errors in the NP-Cloud algorithm, however, in the last lines of page 10 you say: Summing the shifts over the iterative steps yielded a cumulative shift ..., which seems to contradict your response in this instance, that is, there are cumulative iterative calculations. We don't think the error is great, but there is some.

Response: Thank you for this discussion. We realize that “summing the shifts over the iterative steps” is not the best way to describe our algorithm. Instead, each step attempts to correct any remaining error due to the cloud center not being at the origin, resulting in a continued correction-approximation process. An analogy could be made as a missile correcting its path through many adjustments on its path toward the target: the many adjustment steps do not cumulate errors, but instead continue to correct for any residual errors based on the updated position over time. We have revised this text to read “Thus, this stepwise correction process asymptotically shifted the NP-Cloud center to the origin based on the updated position in each iteration, and the total shift after reaching convergence reported on the initial displacement between the two channels.”

clusters

We understand that clusters of localizations are produced in the simulations. In Figure 3c, SMLM cluster FWHM are computed for different algorithms and segment sizes. It is not clear to us what exactly is being shown and how the clusters are determined in the analysis. This needs to be made clearer as the mention here is confusing.

Response: We clarify that in Figure 3c, we are looking at the tropomodulin data, which appear as well-separated clusters due to their ultrastructure in the spectrin-actin cytoskeleton network. We overlaid the drift-corrected single-molecule localizations of different clusters by their centers, and examined their scattering in xy in FWHM. We have added these descriptions to the caption.

3D Data (Z)

You talk about focal locks in commercial microscopes, but, of course, you are excluding those microscopes that do not use focal locks, or make use of z-stacks, or other 3D setups employed in custom-built microscopes. It seems likely that you could expand your algorithm to 3D; 3D is the up and coming trend in microscope design. For sure, you should mention in the abstract that the algorithm is 2D if you are not willing to generalize it, which we understand would require substantial additional work.

Response: Thank you for this discussion. We did not get to address the 3D issue as we tried to address all questions from all reviewers in the limited 3-month timeframe. Now with a focused effort, we are pleased to report that we have accomplished this challenging task. Notably, our algorithms achieved excellent 3D drift correction, substantially outperforming existing methods. As suggested by the reviewer, this new achievement effectively enables 3D-SMLM for systems without focus locks, which we also demonstrate

in our new results. Please find these major updates as new Figure 4, related text, and Supplementary Figs 11 and 12 in this new revision.

typos

Page 5, line 6: ,, -> ,

Page 10, middle: SMLM acquistions. -> SMLM acquisition.

SI Figure 3a and b: "resundant-NPC" in the two bottom left plots in their legends

Response: We thank the reviewer for his/her very careful reading. We have corrected these typos.

Reviewer #1 (Remarks on code availability):

Much improved. The output directory name is: DirftCurveOutput, which has an obvious typo. It would be (as suggested in the original review) nice if the output directory name and/or output filenames somehow identify their data source, so running a different dataset does not overwrite an earlier result from a different source (different path/file).

Documentation is much better and the software is quite usable.

Response: Thank you for noting that our software and documentation are much improved. We have now added different subfolder names for the output of different datasets.

Reviewer #3 (Remarks to the Author):

In this new version of their manuscript, the authors have clarified the various points that I had initially raised.

Reviewer #3 (Remarks on code availability):

I ran the code on simulated and experimental data, which convinced me of the added value of the method.

Response: We thank the reviewer for his/her kind approval.

Reviewer #4 (Remarks to the Author):

This manuscript is improved and most of my concerns have been addressed. I specially appreciate the addition of SI Fig. 8-10 to compare the proposed method to the previous approaches. However, I still believe that the proposed method does not show significant improvement over existing methods and another journal, like Scientific Reports, might be a more suitable place for this paper. It is up to the editors to make this decision.

Response: Thank you for your kind comments. We further note that in this new revision, we have added a major update to extend our approach to achieve excellent 3D drift correction that substantially outperforms existing methods, effectively enabling 3D-SMLM for systems without focus locks.

Response to REVIEWERS' COMMENTS

Reviewer #1 (Remarks to the Author):

The manuscript looks great and ready to publish after fixing a few small typos:

page 4, next to the last paragraph:

also, page 11, new material:

is reported on the initial displacement between the two channels,

Response: Thank you for this discussion. We have revised the sentences in question to “gave an estimation of the initial displacement between the two channels”.

page 9, end of 1st full paragraph

ground-truth drift trajectory, with the in-plan_e_xy shift being identical

Response: Thank you for this discussion. We have corrected this typo.

Reviewer #1 (Remarks on code availability):

All my earlier comments have been addressed.

```

% Load a mat file. Uncomment a line below to load the corresponding file.
% load(fullfile('..', 'SimulatedData', 'SimulatedSMLM_LowDensity10.mat'));
% load(fullfile('..', 'SimulatedData', 'SimulatedSMLM_HighDensity100.mat'));

% load(fullfile('..', 'ExperimentalData', 'TMOD.mat'));
% load(fullfile('..', 'ExperimentalData', 'Spectrin_CTerm.mat'));
% load(fullfile('..', 'ExperimentalData', 'Actin.mat'));
% load(fullfile('..', 'ExperimentalData', 'Microtubules.mat'));
% load(fullfile('..', 'ExperimentalData', 'ER.mat'));
% load(fullfile('..', 'ExperimentalData', 'TOM20.mat'));

% SMLM_Data=0; %Or: paste the data into an array named "SMLM_Data"
% Format: 3 columns single-precision: Frame, X, Y. Assume Frame number
always goes up

%-----Parameters-----
pixelsize = 160; %nm / All test data are based on a pixel size of 160 nm.
All X,Y values are given in pixel unit.

DC_SegmentSize = 70; %Drift-correction segment size: Frames in each segment
for drift correction. Start with 70. Use smaller values for denser samples
and larger values for sparser samples.
MaxSearchRadius_Pix1 = 0.3; %pixel / max search radius btw segments in Pass
1. Start with 0.3. Increase for larger drifts or localization
uncertainties.
MaxSearchRadius_Pix2 = MaxSearchRadius_Pix1 *0.75; %pixel / max search
radius btw segments in Pass 2 is set a bit smaller as the image is already
drift-corrected once and we are refining.
ReSampleFold=12; %Resample fold when compared to the initial count of
molecules in each segment. Start with 12.

%-----
close all;

ArrayFrames=int32(SMLM_Data(:,1));
ArrayX=single(SMLM_Data(:,2));
ArrayY=single(SMLM_Data(:,3));

[Xc,Yc,Xd1,Yd1,Xd2,Yd2, NPC_Time,
RR_NPC_TotalTime]=NPC_RRNP_CallFunction(ArrayX,ArrayY,ArrayFrames,DC_SegmentSize,MaxSearchRadius_Pix1,MaxSearchRadius_Pix2,ReSampleFold)
;

MaxFrameNum=ArrayFrames(end);
DriftFrames=single(1:MaxFrameNum)';
DriftValues1=zeros(MaxFrameNum, 2, 'single');
DriftValues2=zeros(MaxFrameNum, 2, 'single');

DriftValues1(ArrayFrames(1):MaxFrameNum,1:2)=[Xd1 Yd1];
DriftValues2(ArrayFrames(1):MaxFrameNum,1:2)=[Xd2 Yd2];

DriftComb1=[DriftFrames DriftValues1];
DriftComb2=[DriftFrames DriftValues2];

plot(DriftFrames,DriftComb1(:,2),'m',DriftFrames,DriftComb1(:,3),'g',DriftFrames,DriftComb2(:,2),'r
--',DriftFrames,DriftComb2(:,3),'b--')
xlabel('frames');
ylabel('drift (pixel)');
title('NP-Cloud');

%legend('NPC X','NPC Y','RR-NPC X','RR-NPC Y');
legend('NPC X','NPC Y','RR-NPC X','RR-NPC Y', 'Location', 'Best');

%mkdir('DirftCurveOutput')
%OutFileNameHead1=sprintf('DirftCurveOutput/NPC_DriftCurve_prd=%d,DriftRange=%g',DC_SegmentSize,MaxSearchRadius_Pix1)
;
%OutFileNameHead2=sprintf('%s_RR%d',OutFileNameHead1,ReSampleFold);

%saveas(gcf,[OutFileNameHead2 '_drift.png']);
%dlmwrite([OutFileNameHead1,'.txt'], DriftComb1, 'delimiter', '\t',
'precision', 7);
%dlmwrite([OutFileNameHead2,'.txt'], DriftComb2, 'delimiter', '\t',
'precision', 7);

% -----
-----

% David J. Schodt*, Michael J. Wester*, Mohamadreza Fazel, Sajjad Khan,
% Hanieh Mazloom-Farsibaf, Sandeep Pallikkuth, Marjolein B. M. Meddens,
Farzin
% Farzam, Eric A. Burns, William K. Kanagy, Derek A. Rinaldi, Elton Jhamba,
% Sheng Liu, Peter K. Relich, Mark J. Olah, Stanly L. Steinberg and Keith
A.
% Lidke (* = co-1st author), ``SMITE: Single Molecule Imaging Toolbox
% Extraordinaire (MATLAB)', {\sl Journal of Open Source Software}, Volume
8,
% Number 90, 2023, p. 5563,
% {\sf https://joss.theoj.org/papers/10.21105/joss.05563},
% (DOI: 10.21105/joss.05563).

DC = smi_core.DriftCorrection;
NFramesPerDataset = 100;

n_frames = max(SMLM_Data(:, 1));
% Make n_frames divisible by NFramesPerDataset.
n_frames = floor(n_frames / NFramesPerDataset) * NFramesPerDataset;

```

```

% Remove ignored frames.
idx = find(SMLM_Data(:, 1) > n_frames);
SMLM_Data_a = SMLM_Data;
SMLM_Data_a(idx, :) = [];

SMDin.NFrames = n_frames;
SMDin.FrameNum = SMLM_Data_a(:, 1);
SMDin.X = SMLM_Data_a(:, 2);
SMDin.Y = SMLM_Data_a(:, 3);
n_loc = numel(SMDin.X);
SMDin.NDatasets = 1;
SMDin.DatasetNum = repmat(1, n_loc, 1);

% Intra-dataset threshold (pixel)
%DC.L_intra = 1;
% Inter-dataset threshold (pixel)
%DC.L_inter = 2;
% X/Y pixel size in um (only needed for 3D drift correction)
%DC.PixelSizeZUnit = 0.1;
% Degree of the intra-dataset fitting polynomial for drift rate
%DC.PDegree = 1;
% Termination tolerance on the intra-dataset function value
%DC.TolFun_intra = 1e-2;
% Termination tolerance on the intra-dataset fitting polynomial
%DC.TolX_intra = 1e-4;
% Termination tolerance on the inter-dataset function value
%DC.TolFun_inter = 1e-2;
% Termination tolerance on the inter-dataset fitting polynomial
%DC.TolX_inter = 1e-4;
% Initialization wrt the previous dataset for inter-dataset drift
correction
% The value should be either 0 (no initial drift), 1 (initial drift of the
% previous dataset) or SMD.NFrames (final drift); zero or initial drift
% should work well with brightfield registration, while final drift works
% well generally (but the optimization process may not converge quite as
% quickly).
DC.Init_inter = 1;
% If non-empty, override the collected value of number of datasets
DC.NDatasets = [];
% If non-empty, override the collected value of number of frames per
dataset
DC.NFrames = NFramesPerDataset;
% Verbosity level
%DC.Verbose = 1;

[SMDout, Statistics] = DC.driftCorrectKNN(SMDin);
Statistics

figure(2)
hold on
plot(1 : SMDout.NFrames, SMDout.DriftX);
plot(1 : SMDout.NFrames, SMDout.DriftY);
legend('X', 'Y', 'Location', 'Best');
xlabel('frames');
ylabel('drift (pixel)');
title('driftCorrectKNN');
hold off

```